# EXOC1 plays an integral role in spermatogonia pseudopod elongation and spermatocyte stable syncytium formation in mice

Yuki Osawa[1†], Kazuya Murata[2†], Miho Usui[3], Yumeno Kuba[1], Hoai Thu Le[4], Natsuki Mikami[4], Toshinori Nakagawa[5,6], Yoko Daitoku[2], Kanako Kato[2], Hossam Hassan Shawki[7], Yoshihisa Ikeda[8], Akihiro Kuno[2,4], Kento Morimoto[9], Yoko Tanimoto[2], Tra Thi Huong Dinh[2], Ken-ichi Yagami[2], Masatsugu Ema[10], Shosei Yoshida[5,6], Satoru Takahashi[2], Seiya Mizuno[2*], Fumihiro Sugiyama[2]

[1]Master's Program in Medical Sciences, Graduate School of Comprehensive Human Sciences, University of Tsukuba, Tsukuba, Japan; [2]Laboratory Animal Resource Center, Trans-border Medical Research Center, University of Tsukuba, Tsukuba, Japan; [3]School of Medical Sciences, University of Tsukuba, Tsukuba, Japan; [4]Ph.D Program in Human Biology, School of Integrative and Global Majors, University of Tsukuba, Tsukuba, Japan; [5]Division of Germ Cell Biology, National Institute for Basic Biology, National Institutes of Natural Sciences, Okazaki, Japan; [6]Department of Basic Biology, School of Life Science, Graduate University for Advanced Studies (Sokendai), Okazaki, Japan; [7]Department of Comparative and Experimental Medicine, Nagoya City University Graduate School of Medical Sciences, Nagoya, Japan; [8]Doctoral program in Biomedical Sciences, Graduate School of Comprehensive Human Sciences, University of Tsukuba, Tsukuba, Japan; [9]Doctoral program in Medical Sciences, Graduate School of Comprehensive Human Sciences, University of Tsukuba, Tsukuba, Japan; [10]Department of Stem Cells and Human Disease Models, Research Center for Animal Life Science, Shiga University of Medical Science, Otsu, Japan

**\*For correspondence:**
konezumi@md.tsukuba.ac.jp

[†]These authors contributed equally to this work

**Competing interests:** The authors declare that no competing interests exist.

**Abstract** The male germ cells must adopt the correct morphology at each differentiation stage for proper spermatogenesis. The spermatogonia regulates its differentiation state by its own migration. The male germ cells differentiate and mature with the formation of syncytia, failure of forming the appropriate syncytia results in the arrest at the spermatocyte stage. However, the detailed molecular mechanisms of male germ cell morphological regulation are unknown. Here, we found that EXOC1, a member of the Exocyst complex, is important for the pseudopod formation of spermatogonia and spermatocyte syncytia in mice. EXOC1 contributes to the pseudopod formation of spermatogonia by inactivating the Rho family small GTPase Rac1 and also functions in the spermatocyte syncytia with the SNARE proteins STX2 and SNAP23. Since EXOC1 is known to bind to several cell morphogenesis factors, this study is expected to be the starting point for the discovery of many morphological regulators of male germ cells.

## Introduction

In mammalian testis, sperms are continuously produced in the seminiferous tubules throughout a male's life. There are two types of cells in the seminiferous tubule, somatic cells (Sertoli cells) which

support spermatogenesis and male germ cells which undergo various processes to become spermatogonia, spermatocytes, spermatids, and eventually spermatozoa. Structurally, the Sertoli cell tight junction (SCTJ) divides the seminiferous tubule into two compartments, the basal compartment and the luminal compartment (*Tsukita et al., 2008*). Spermatogonia cells are located in the basal compartment and undergo proliferation and differentiation by mitosis. The cells that have differentiated into spermatocytes migrate to the luminal compartment and differentiate into haploid spermatocytes by meiosis (*Smith and Braun, 2012*). These spermatocytes then undergo dynamic morphological changes to become mature sperms that are released into the lumen of the seminiferous tubule. As this entire differentiation process is cyclical and continuous, long-term stable spermatogenesis is supported by spermatogonial stem cells.

Murine spermatogonia are heterogeneous in their differential state and can be classified as undifferentiated and differentiating (*Fayomi and Orwig, 2018*; *Suzuki et al., 2009*). During spermatogenesis, germ cells undergo an incomplete cytoplasmic division. In most somatic cell types, daughter cells completely separate from each other, but in germ cells, the final separation is inhibited and they differentiate while maintaining intercellular bridges (ICBs) (*Iwamori et al., 2010*). Such structures, called syncytia, are thought to contribute to syncytial differentiation and maturation by the sharing of mRNAs and proteins that are halved during meiosis via ICB (*Morales et al., 2002*; *Fawcett et al., 1959*). Undifferentiated spermatogonia are a morphologically heterogeneous population that includes $A_{single}$ as isolated cells, $A_{pair}$ as two connected syncytia, and $A_{aligned}$ as three or more connected cells. Furthermore, a large number of stem cells that contribute to long-term spermatogenesis are included within the $GFR\alpha1^+$ undifferentiated spermatogonia (mainly $A_{single}$ and $A_{pair}$, and fewer $A_{aligned}$), although whether the stem cell compartment comprises of all or only a subset of $GFR\alpha1^+$ population remains under debate (*Lord and Oatley, 2017*). The $GFR\alpha1^+$ cells produce $NGN3^+/RAR\gamma^+$ subset of undifferentiated spermatogonia (composed of more $A_{aligned}$, and fewer $A_{single}$ and $A_{pair}$) (*Nakagawa et al., 2007*) that differentiate into differentiating spermatogonia ($Kit^+$) in response to retinoic acid. These undifferentiated spermatogonia move within the basal compartment of the seminiferous tubules and regulate the balance between self-renewal and differentiation by competing for fibroblast growth factors (FGFs) that are secreted from deferent niches (*Kitadate et al., 2019*). Although transplantation studies using in vitro cultured spermatogonial stem cells have reported that Rac1-mediated cell migration is important for spermatogonial stem cell homing (*Takashima et al., 2011*), the molecules that are involved in spermatogonia migration and their mechanisms have not been clarified (*Kanamori et al., 2019*).

TEX14 is an important molecule for stability of ICBs, which are essential for maintaining the structure of syncytia. This protein localizes to germline ICBs and inhibits the final separation of the cytoplasm; in mice lacking this gene, the cytoplasm of male germ cells is completely separated causing spermatogenesis to be halted (*Greenbaum et al., 2006*). This result indicates that the formation of syncytium is essential for spermatogenesis. Seminolipids, which are sulfated glycolipids, are important for the sustenance of ICB in spermatocytes. This is evident in mice lacking the *Ugt8a* and *Gal3st1*, required for seminolipids synthesis, as they show an aggregation of spermatocytes (*Fujimoto et al., 2000*; *Honke et al., 2002*). Although the detailed molecular mechanism is unknown, a similar phenotype was observed in mice lacking the *Stx2* gene (*Fujiwara et al., 2013*), which functions in the transport of seminolipids to the cell membrane. In humans, similar to that of mice, the loss of function of *STX2* results in the failure of spermatogenesis with spermatocyte aggregation (*Nakamura et al., 2018*).

In this study, we focus on the exocyst complex, a heterodimeric protein complex composed of eight subunits (EXOC1–EXOC8) that are widely conserved from yeast to humans (*Koumandou et al., 2007*). Exocyst-mediated vesicle trafficking is crucial for various fundamental cellular phenomena such as cell division, morphogenesis, and cell migration. In cytokinesis, the exocyst is localized to ICB (*Neto et al., 2013*) and is required for the transport of factors required for the final separation of the plasma membrane (*Kumar et al., 2019*). The exocyst has also been implicated in cell migration via the reconstitution of the actin cytoskeleton (*Liu et al., 2012*; *Parrini et al., 2011*).

Although studies using gene-deficient mouse models have shown that the exocyst is important for embryo development (*Mizuno et al., 2015*; *Friedrich et al., 1997*; *Fogelgren et al., 2015*; *Dickinson et al., 2016*), its tissue-specific functions in adults are largely unknown. In *Drosophila* models, the exocyst plays an important role in the formation of both female and male gametes

(*Giansanti et al., 2015*; *Wan et al., 2019*; *Murthy and Schwarz, 2004*; *Mao et al., 2019*). In mice, all exocyst subunits are expressed in male germ cells at each stage of differentiation (*Green et al., 2018*), and although it is predicted that these subunits may be involved in mammalian spermatogenesis, the function of these subunits is completely unknown. Therefore, in this study, we generated and analyzed a mouse model deficient in *Exoc1*, an exocyst subunit expressed in male germ cells.

## Results

### EXOC1 expression in mouse testis

The expression of *Exoc1* mRNA has been observed in Sertoli cells and the germ cells of mouse testes at each stage: spermatogonia, spermatocytes, round spermatids, and elongating spermatids (*Green et al., 2018*; *Figure 1—figure supplement 1A* and *Figure 1—figure supplement 2*). To analyze the expression of EXOC1 at the protein level, we generated two types of genome-edited mice in which the *PA-Tag* sequence was knocked in at the C-terminus of *Exoc1* (*Exoc1$^{PA-C}$*) or the N-terminus (*Exoc1$^{PA-N}$*) using the CRISPR-Cas system (*Figure 1A*) as immunofluorescence using commercially available antibodies for EXOC1 failed to detect any signal. The LG3 sequence, which is a flexible linker, was used as there was no successful case for producing a knock-in (KI) homozygous mutant implying that the functionality of EXOC1 has not been lost (*Ahmed et al., 2018*). In both strains, mice with the intended KI sequences were obtained (*Figure 1—figure supplement 1B*), and both heterozygous mutants showed no abnormal phenotypes. We attempted to produce homozygous mutants by intercrossing the hetero KI mutants of each strain. In the *Exoc1$^{PA-C}$* strain, no homozygous mutants were obtained (0/20). By contrast, in the *Exoc1$^{PA-N}$* strain, homozygous mutants were obtained (9/27), and no abnormal phenotypes were found in them. Furthermore, progenies were obtained from crosses between *Exoc1$^{PA-N}$* homozygous mutants. As the *Exoc1* knock-out (KO) mice are early embryonic lethal (*Mizuno et al., 2015*), this result suggests that the addition of the PA-tag connected to the N-terminus, linked by LG3, does not impair the function of EXOC1. We

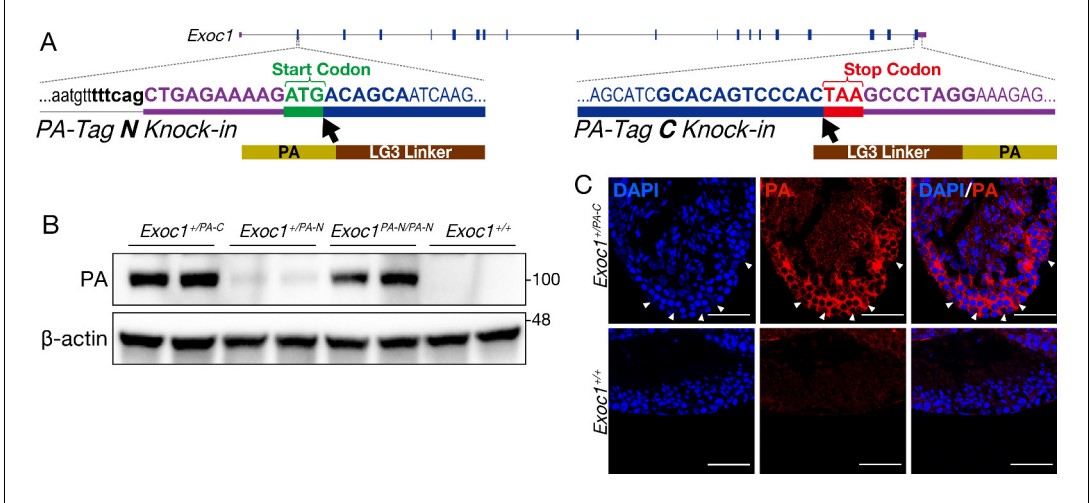

**Figure 1.** Confirmation of EXOC1 expression in testes using the PA-Tag knock-in mouse. (A) Generation of PA-Tag knock-in mice. In the *Exoc1$^{PA-C}$* allele, the LG3-linker connected PA-tag gene fragment was knocked-in just before the stop codon of *Exoc1* using CRISPR-Cas9. In the *Exoc1$^{PA-N}$* allele, the LG3-connected PA-tag gene fragment was knocked in just after the start codon of *Exoc1* using CRISPR-Cas12a. Bold letters represent the CRISPR-Cas9 and Cas12a target sequence. (B) Western blotting of PA-Tag antibody demonstrated that each C- and N-terminal PA-tagged EXOC1 protein was expressed in the adult testes (n = 2 in each genotype). (C) Immunofluorescence with PA-Tag antibody. EXOC1 is observed in every cell in the adult testes. The arrowheads indicate Sertoli cells in which the nucleus is eurochromatin with a large nucleolus. Scale bars: 50 μm.

The online version of this article includes the following source data and figure supplement(s) for figure 1:

**Source data 1.** Raw data of the in vivo western blot.

**Figure supplement 1.** Expression of mRNA for each Exocyst subunit and EXOC1 protein in adult mouse male germ cells.

**Figure supplement 2.** Expression of Exocyst subunits in Sertoli cells of adult mice.

**Figure supplement 3.** The production of *Exoc1* flox mice.

performed western blot analyses for the PA-tag in the testes of both $Exoc1^{PA-C}$ and $Exoc1^{PA-N}$ adult (10–20-week-old) mice and confirmed a band of the molecular size as expected (*Figure 1B* and *Figure 1—source data 1*). The signal intensity of the band, on the western blot, was lower for $Exoc1^{+/PA-N}$ than that for $Exoc1^{+/PA-C}$. This intensity difference may be due to differences in their kinetics, structure, subcellular localization, and so on. Since the $Exoc1^{PA-N}$ homozygous mutant, unlike the $Exoc1^{PA-C}$ homozygous mutant, did not show a pronounced abnormal phenotype, we considered that EXOC1-PA-N is probably more similar in behavior and function to the wild-type EXOC1 than EXOC1-PA-C. In immunofluorescence, EXOC1-PA was also detected in all male germ cells observed in the $Exoc1^{+/PA-C}$ adult mice (*Figure 1C* and *Figure 1—figure supplement 1C*), while no signal could be detected in the $Exoc1^{PA-N/PA-N}$ mice using any method. In $Exoc1^{+/PA-C}$ adult mice, PA signals were also detected in Sertoli cells, which are located at the basal compartment of the seminiferous tubules and whose nucleus are euchromatic with a large nucleolus (*França et al., 2016*). These data indicate that EXOC1 protein is expressed in male mouse germ and Sertoli cells.

## Impaired spermatogenesis in *Exoc1* conditional KO mice

*Exoc1* deficiency results in peri-implantation embryonic lethality in mice (*Mizuno et al., 2015*). In this study, we analyzed the function of *Exoc1* in spermatogenesis using *Exoc1* conditional knockout (cKO) mice obtained by crossing *Exoc1*-floxed mice (*Figure 1—figure supplement 3*) with *Nanos3-Cre* driver that expresses *Cre* in spermatogonia (*Suzuki et al., 2008*). To investigate the spermatogenic potential of *Exoc1* cKO mice, PNA-lectin staining to detect acrosome was performed; however, no signal was detected in the *Exoc1* cKO adult mice (*Figure 2A*). When cell morphology was observed by H and E staining, although almost no spermatids and spermatozoa were observed in the testes of *Exoc1* cKO mice, large multinucleated cells were observed in the lumen of seminiferous tubules (*Figure 2B and C*). During spermatogenesis, male germ cells form syncytia and are interconnected via stable ICB (*Iwamori et al., 2010*). Detailed observations by scanning electron microscopy revealed ICB in control mice, while multinucleated cells that lacked ICB structures were observed in *Exoc1* cKO mice (*Figure 2D*). As the observed multinucleated cells appeared to be aggregates of syncytia, they were named 'AGS' for the purpose of this study. To confirm the frequency of AGS in the *Exoc1* cKO adult mice, H and E-stained sections of the seminiferous tubules were classified into three categories: sections containing neither AGS nor sperm, sections containing AGS, and sections containing sperm but no AGS; and counted (*Figure 2—figure supplement 1A*). More than 70% of the sections contained AGS, and approximately 20% contained neither AGS nor sperm (*Figure 2—figure supplement 1B and C*). Although less than 10% of the cross-sections contained sperms, these sperms were considered to be derived from spermatogonia that did not undergo Cre-LoxP recombination. We confirmed the genotype of the generated blastocysts by in vitro fertilization of wild-type oocytes with sperms from the *Exoc1* cKO or control ($Exoc1^{flox/-}$) mice and found embryos with the cKO allele in the control group but not in the *Exoc1* cKO group (0/16) (*Figure 2—figure supplement 1D*). To determine at which stage of differentiation the AGS emerge from, immunofluorescence using male germ cell markers for each differentiation stage were used. γH2AX, a spermatocyte marker for the pachytene stage, was observed in the nuclei of AGS (*Figure 2E*). In contrast, no aggregation was observed in the differentiating spermatogonia, Kit$^+$ syncytia (0/214) (*Figure 2F* and *Supplementary file 1a*). As the seminiferous tubules are divided into two compartments by the SCTJ, spermatogonia located within the basal compartment migrate to the luminal compartment when they become spermatocytes (*Smith and Braun, 2012*). In order to investigate the arrangement of AGS in the seminiferous tubules, H and E staining and immunofluorescence for CLDN11, a major component of SCTJ (*Morita et al., 1999*), were performed, and all of the 336 AGS in *Exoc1* cKO adult mice were observed within the luminal compartment (*Figure 2G*). These data suggest that the AGS seen in *Exoc1* cKO mice are spermatocytes.

In the pachytene stage, chromosome synapsis occurs and the synaptonemal complex is formed (*Heyting, 1996*). To confirm this synapsis, chromosomes of spermatocytes from *Exoc1* adult cKO mice were spread and immunofluorescence was performed using antibodies against SYCP1 and SYCP3, which are components of the synaptonemal complex. SYCP1 and SYCP3 signals were colocalized in autosomal pairs of the spermatocytes from *Exoc1* cKO mice. These results suggest that meiosis proceeds normally to synapsis in *Exoc1*-deficient spermatocytes (*Figure 2—figure supplement 2*).

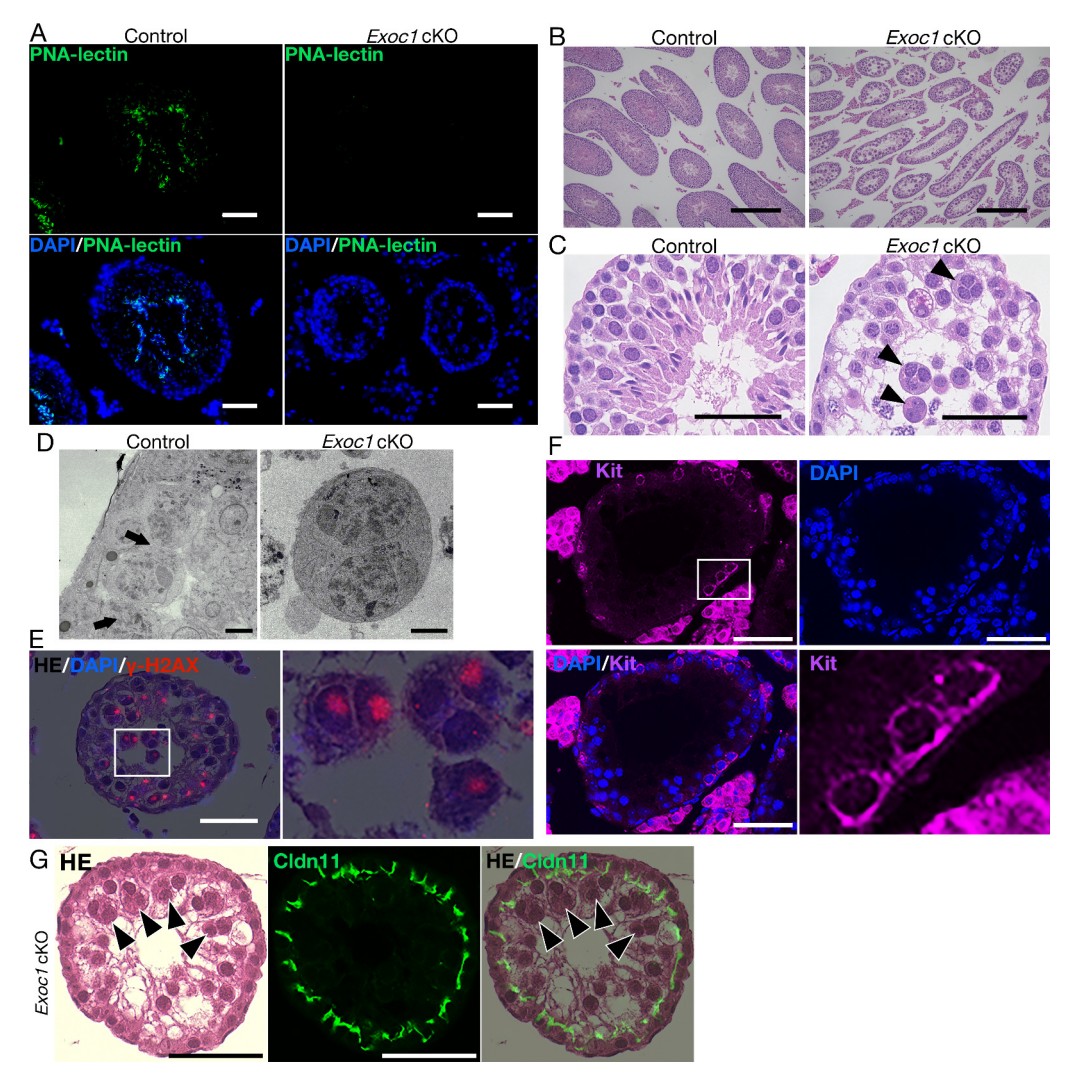

**Figure 2.** Impaired spermatogenesis in *Exoc1* cKO mice. (**A**) PNA-lectin staining of the *Exoc1* adult cKO testis. Signals of PNA-lectin, an acrosomal marker was not observed in the *Exoc1* cKO testis. Scale bars: 50 μm. (**B**) The macroscopic images for H and E staining. Normal spermatogenesis was not observed in almost all of the seminiferous tubules in the *Exoc1* cKO adult testis. Scale bars: 300 μm. (**C**) The mesoscopic images for H and E staining of the *Exoc1* cKO testis. Large and circular cells, appearing to be aggregates of syncytia (AGS), containing multiple nuclei were observed in the lumen of seminiferous tubules (arrowheads). Scale bars: 100 μm. Control: *Exoc1^{flox/wt}:: Nanos3^{+/Cre}* adult mice. (**D**) SEM observation of the *Exoc1* cKO adult testis. Intercellular bridges (ICBs) (arrows) were found in the syncytia in the control (*Exoc1^{flox/wt}:: Nanos3^{+/Cre}*) testis. There were no ICB observed in the AGS of *Exoc1* cKO. Scale bars: 5 μm. (**E**) The serial section overlay image of *Exoc1* cKO adult testis. There were γ-H2AX (marker of spermatocyte) signals in the AGS nucleus. Scale bars: 50 μm. (**F**) A representative immunofluorescence image of an *Exoc1* cKO seminiferous tubule. Kit^{+} syncytia, which are observed in differentiating spermatogonia, have ICB. Scale bars: 50 μm. (**G**) H and E staining and immunofluorescence with CLDN11. CLDN11-positive Sertoli cell tight junction (SCTJ) divides the space between the basal and the luminal compartment, and AGS are present within the luminal compartment (arrowheads). Scale bars: 50 μm.

The online version of this article includes the following figure supplement(s) for figure 2:

**Figure supplement 1.** The occurrence frequency of aggregates of syncytia (AGS) in the *Exoc1* cKO.

**Figure supplement 2.** Normal meiotic chromosome synapsis in *Exoc1* cKO mice.

## EXOC1 regulates ICB formation in cooperation with STX2 and SNAP23

The spermatocytes of *Stx2^{repro34}* mice, a null mutant of the *Stx2* gene encoding the SNARE protein, exhibit multinucleated AGS (*Fujiwara et al., 2013*). Both *Stx2^{repro34}* and *Exoc1* cKO mice show similar phenotypes of multinucleated AGS and normal chromosome synapsis (*Fujiwara et al., 2013*). In addition, in yeast, Sec3 (an ortholog of mouse *Exoc1*) promotes membrane fusion between transport

vesicles and the cell membrane by coordinating with Sso2 (an ortholog of mouse *Stx2*) and another SNARE protein, Sec9 (an ortholog of mouse *Snap23*) (*Yue et al., 2017*). Therefore, we hypothesized that EXOC1 is required for normal syncytia formation in mouse spermatocytes to function in cooperation with STX2 and SNAP23. In yeast, Sec3 directly interacts with Sso2 and promotes the Sso2–Sec9 binary complex formation (*Yue et al., 2017*). As there were no previous reports regarding the interaction between EXOC1 and STX2 in mice, we analyzed the binding of EXOC1-STX2 and STX2-SNAP23 by the co-immunoprecipitation (Co-IP) of HEK293T cells overexpressing EXOC1, STX2, and SNAP23 from mice with FLAG, HA, and MYC epitope tags, respectively. The binding of the three proteins was confirmed in all in vitro experiments where each protein was immunoprecipitated (*Figure 3A* and *Figure 3—source data 1*). We then confirmed the binding of SNAP23 to EXOC1 in the adult *Exoc1$^{PA-N/PA-N}$* testis with anti-Snap23 antibody available for immunoprecipitation. Consequently, it was confirmed that EXOC1 binds to SNAP23 in vivo (*Figure 3B* and *Figure 3—source data 2*). These results, consistent with those reported in yeast (*Yue et al., 2017*), suggested that mouse EXOC1 could bind to STX2 and STX2 to SNAP23.

The Co-IP described above confirmed that EXOC1, STX2, and SNAP23 interact in mice, and given the hypothesis that EXOC1 functions in cooperation with STX2 and SNAP23 to form normal syncytium of spermatocytes. To test this hypothesis, we conducted a functional analysis of SNAP23. As *Snap23* KO mice have been reported to be embryonically lethal (*Suh et al., 2011*) and the effect of *Snap23* depletion on adult male germ cells was unknown, we generated *Snap23* cKO mice specifically for male germ cells under the control of *Nanos3* expression (*Figure 3—figure supplement 1*) and tested for the appearance of spermatocyte AGS in the adult testes. H and E staining and immunofluorescence showed that AGS with γH2AX signals were observed in *Snap23* cKO mice (*Figure 3C–E*). However, as many spermatocytes did not aggregate in *Snap23* cKO mice, spermatozoa were also observed by H and E and PNA-lectin staining (*Figure 3D and F*). We evaluated the appearance of AGS in *Snap23* cKO adult mice by the same method as in the *Exoc1* cKO mice and found sperms in more than 70% of the seminiferous tubule sections (*Figure 3—figure supplement 2A and B*). Although AGS were observed in about 10% of the sections, the number of AGS per area in those sections was approximately 10% of that in the *Exoc1* cKO mice (*Figure 3—figure supplement 2C* and *Figure 3—source data 3*). This suggests that *Snap23* is dispensable for spermatogenesis and that another protein in the SNAP family could be compensating for that function. Alternatively, although the analyzed ages, floxed distances (*Figure 1—figure supplement 3* and *Figure 3—figure supplement 1*), and Cre driver strains were almost exactly the same, the possibility that this phenotypic difference was due to Cre-LoxP recombination efficiency could not be completely excluded. In any case, *Snap23* is partially responsible for the formation or maintenance of the spermatocyte syncytium. These results suggest that EXOC1 regulates the formation of the correct syncytium structure in cooperation with STX2 and SNAP23.

## EXOC1 regulates the pseudopod elongation via Rac1 inactivation

As *Nanos3*-Cre mice express CRE from spermatogonia (*Suzuki et al., 2008*), the *Exoc1* cKO (*Exoc1-$^{Flox/Flox}$:: Nanos3$^{+/Cre}$*) mice used in this study were models in which the *Exoc1* gene was deficient from spermatogonia. Therefore, we investigated the effects of *Exoc1* deficiency in spermatogonia. In a study of cultured cells, the exocyst complex was noted to contribute to cell migration by promoting actin assembly (*Liu et al., 2012*) and GFRα1$^+$ undifferentiated spermatogonia were observed to be moving within the basal compartment of the seminiferous tubules (*Hara et al., 2014*). For these reasons, we carried out detailed morphological observations of GFRα1$^+$ spermatogonia using confocal microscopy. The pseudopods of GFRα1$^+$ spermatogonia in *Exoc1* cKO adult mice were significantly shorter than that of control mice (*Figure 4A*). In the formation of syncytia in spermatocytes, *Exoc1* is likely to function in concert with *Stx2* (*Figure 3*). In contrast, as there are no reports on the function of *Stx2* in the formation of pseudopods in spermatogonia, it is unclear whether *Exoc1* cooperates with *Stx2* in this phenomenon as well. Therefore, we generated *Stx2* KO mice using the CRISPR-Cas9 system (*Figure 4—figure supplement 1A and B*) and performed detailed morphological observations of GFRα1$^+$ spermatogonia. In the *Stx2* KO mice generated in this study, as in the *Stx2$^{repro34}$* mice (*Fujiwara et al., 2013*), spermatocytes were observed to aggregate (*Figure 4—figure supplement 1C*). Pseudopod length was quantitatively assessed by Sholl analysis: GFRα1$^+$ spermatogonia in *Exoc1* cKO mice had significantly shorter pseudopods than those of wild-type and *Stx2* KO (*Figure 4B* and *Figure 4—source data 1*). The *Stx2* KO pseudopods tended to be shorter

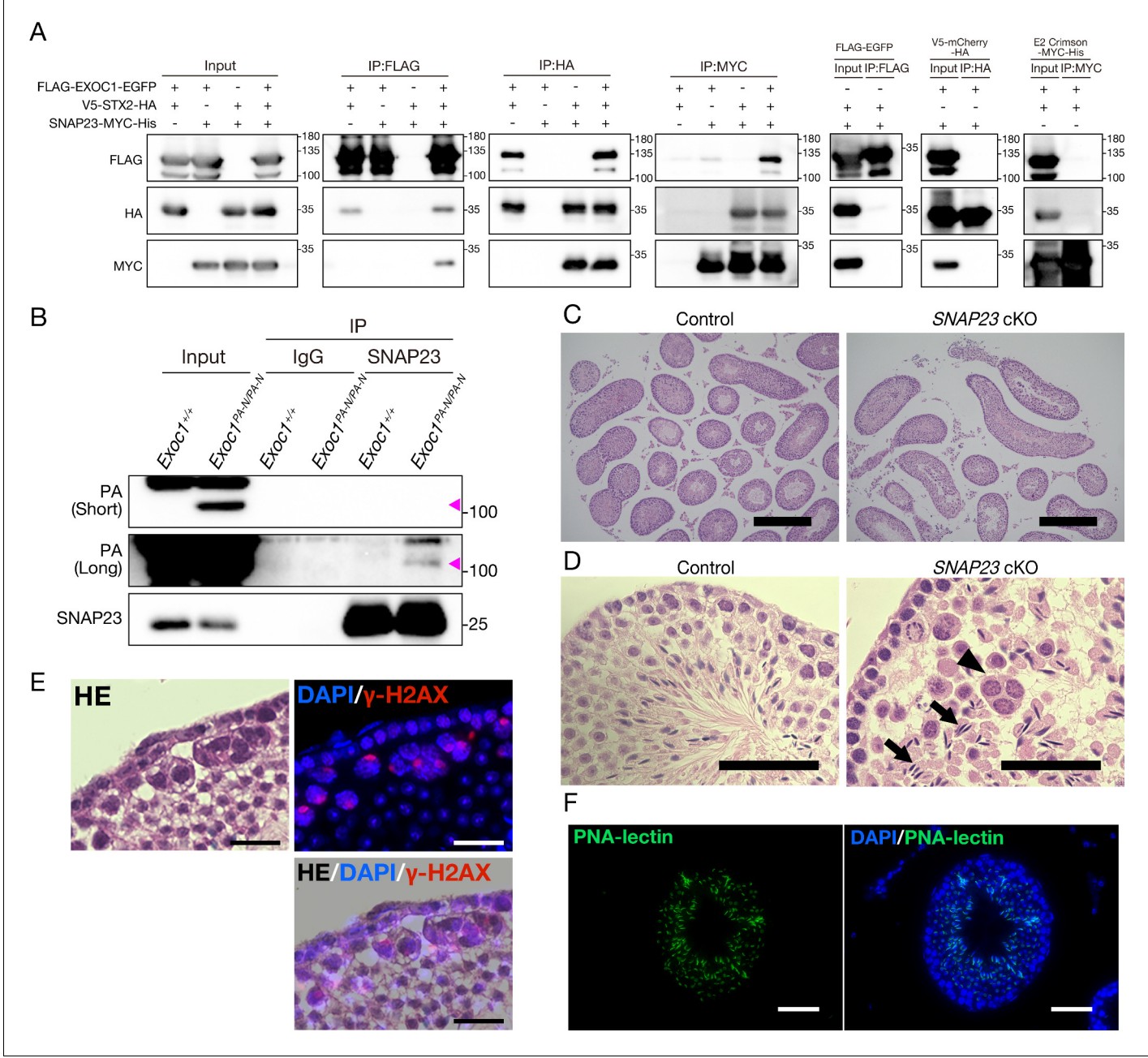

**Figure 3.** EXOC1 regulates ICB formation in cooperation with STX2 and SNAP23. (**A**) Co-immunoprecipitation of EXOC1-STX2-SNAP23 complex in vitro. FLAG-tagged mouse EXOC1, HA-tagged mouse STX2, and Myc-tagged mouse SNAP23 were co-overexpressed in HEK293T cells. The binding of the three factors was confirmed in all combinations of Co-IP experiments. FLAG-EGFP, V5-mCherry-HA, and E2 crimson-MYC-His were used as negative controls. (**B**) Interaction of EXOC1 with SNAP23 in vivo. PA-tagged EXOC1 was co-immunoprecipitated with endogenous SNAP23 in the adult *Exoc1*<sup>PA-N</sup> testis. The upper and middle panels show short and long period exposure images, respectively. Arrowhead indicates PA-EXOC1. (**C**) The macroscopic images for H and E staining. Sperms were found in frequent seminiferous tubules in adult *Snap23* cKO mice. Scale bars: 300 μm. Control: *Snap23*<sup>flox/wt</sup>:: *Nanos3*<sup>+/Cre</sup> mice. (**D**) The mesoscopic images for H and E staining of *Snap23* cKO adult testis. Large and circular cells containing multiple nuclei (arrowheads), appearing to be aggregates of syncytia (AGS) were observed. In contrast with *Exoc1* cKO, every seminiferous tubule had sperms with elongated nuclei (arrows). Scale bars: 50 μm. Control: *Snap23*<sup>flox/wt</sup>:: *Nanos3*<sup>+/Cre</sup> mice. (**E**) The serial section overlay image of *Snap23* cKO testis. Immunofluorescence signals of γ-H2AX were found in AGS. Scale bars: 50 μm. (**F**) PNA-lectin staining of *Snap23* cKO testis. PNA-lectin that was used to detect the acrosome is observed in the lumen of the seminiferous tubule of *Snap23* cKO testis. Scale bars: 50 μm.

The online version of this article includes the following source data and figure supplement(s) for figure 3:

**Source data 1.** Raw data of the in vitro immunoprecipitation.
**Source data 2.** Raw data of the in vivo immunoprecipitation.

*Figure 3 continued on next page*

*Figure 3 continued*

**Source data 3.** Occurrence rate of AGS per area in the extracted Section containing AGS.
**Figure supplement 1.** The production of *Snap23* flox mice.
**Figure supplement 2.** The occurrence frequency of aggregates of syncytia (AGS) in the *Snap23* cKO.

than wild-type pseudopods (p=0.052) (*Figure 4A and B* and *Figure 4—source data 1*). These results suggest that EXOC1 functions in the pseudopod elongation of GFRα1$^+$ spermatogonia partially dependently, but not completely dependently of STX2. The GFRα1$^+$ cell population, which should be a component of the spermatogonial stem cell pool, is mostly A$_{single}$ and A$_{pair}$, with a minority of A$_{aligned}$ (*Hara et al., 2014*). The elongation of the pseudopod in each of these morphological states was confirmed through whole-mount immunofluorescence staining with adult *Exoc1* cKO mice (*Figure 4C*). In this experiment, pseudopod elongation was inhibited in GFRα1$^+$ A$_{single}$ spermatogonia of the *Exoc1* cKO group compared to the control group (*Figure 4D* and *Figure 4—source data 2*). The distance between the connected cells of A$_{pair}$ and A$_{aligned}$ was significantly shorter in GFRα1$^+$ A$_{pair}$ spermatogonia in the *Exoc1* cKO group than in the control group (*Figure 4E and F* and *Figure 4—source data 2*).

We examined the molecular mechanism by which EXOC1 functions in GFRα1$^+$ spermatogonia pseudopod elongation. Although the exocyst contributes to pseudopod formation by promoting Arp2/3-mediated actin assembly (*Liu et al., 2012*), Arp3 is not expressed in spermatogonia (*Lie et al., 2010*). Thus, we focused on the Rho family GTPase Rac1, which regulates pseudopod elongation and cell migration by regulating the reconstruction of the actin cytoskeleton (*Raftopoulou and Hall, 2004*). The exocyst is required for the transport of SH3BP1, which converts Rac1 from its active to inactive form, and when the exocyst is impaired in cultured human cells, Rac1 is over-activated and pseudopod elongation and cell migration are inhibited (*Parrini et al., 2011*). Therefore, immunofluorescence for active-Rac1 was performed to examine Rac1 activity in GFRα1$^+$ spermatogonia of *Exoc1* cKO adult mice, and as expected, the fluorescence intensity of active-Rac1 tended to be stronger for the plasma membrane of GFRα1$^+$ spermatogonia of *Exoc1* cKO mice compared to control mice (*Figure 4G and H*, and *Figure 4—source data 3*). Therefore, EXOC1 may function in GFRα1$^+$ spermatogonia pseudopod elongation by negatively regulating the activity of Rac1.

## Exoc1 functions to regulate the differentiation of spermatogonia

GFRα1$^+$ cell migration dictates the state of spermatogonia differentiation (*Kitadate et al., 2019*). Therefore, we counted the numbers of GFRα1$^+$ cells and RARγ$^+$ cells, which are one differentiation step from GFRα1$^+$ cells (*Kitadate et al., 2019*). Immunofluorescence with antibodies to GFRα1 and RARγ showed a significant increase in the number of GFRα1$^+$ cells and a significant decrease in the number of RARγ$^+$ cells in *Exoc1* cKO adult mice compared to the control mice (*Figure 5A and B* and *Figure 5—source data 1*). In addition, no noticeable difference in the total number of both cells (*Figure 5B*). Additionally, the number of Kit$^+$ differentiating spermatogonia significantly decreased in the *Exoc1* cKO group (*Figure 5—figure supplement 1* and *Figure 5—source data 2*). Taken together, these results suggest that *Exoc1* strongly regulates spermatogonial differentiation but not proliferation as much.

## Discussion

This study is the first to show that EXOC1 plays an essential role in spermatogenesis, in particular the formation of germ cell morphology, in the elongation of the pseudopod in spermatogonia, and in the formation of syncytium in spermatocytes.

*Exoc1* cKO mice showed overactivity of Rac1 and disturbance of differentiation balance in spermatogonia. However, the presence of the next stage of differentiation, spermatocytes, suggests that abnormalities in the spermatogonia are not a direct factor in the failure of spermatogenesis exhibited by the *Exoc1* cKO mice. In contrast, as spermatocytes failed to maintain ICB and aggregated, there were no germ cells observed after spermatid stage. Since chromosome synapsis is suggested to be normal in meiosis, the most significant factor in spermatogenesis failure may be the

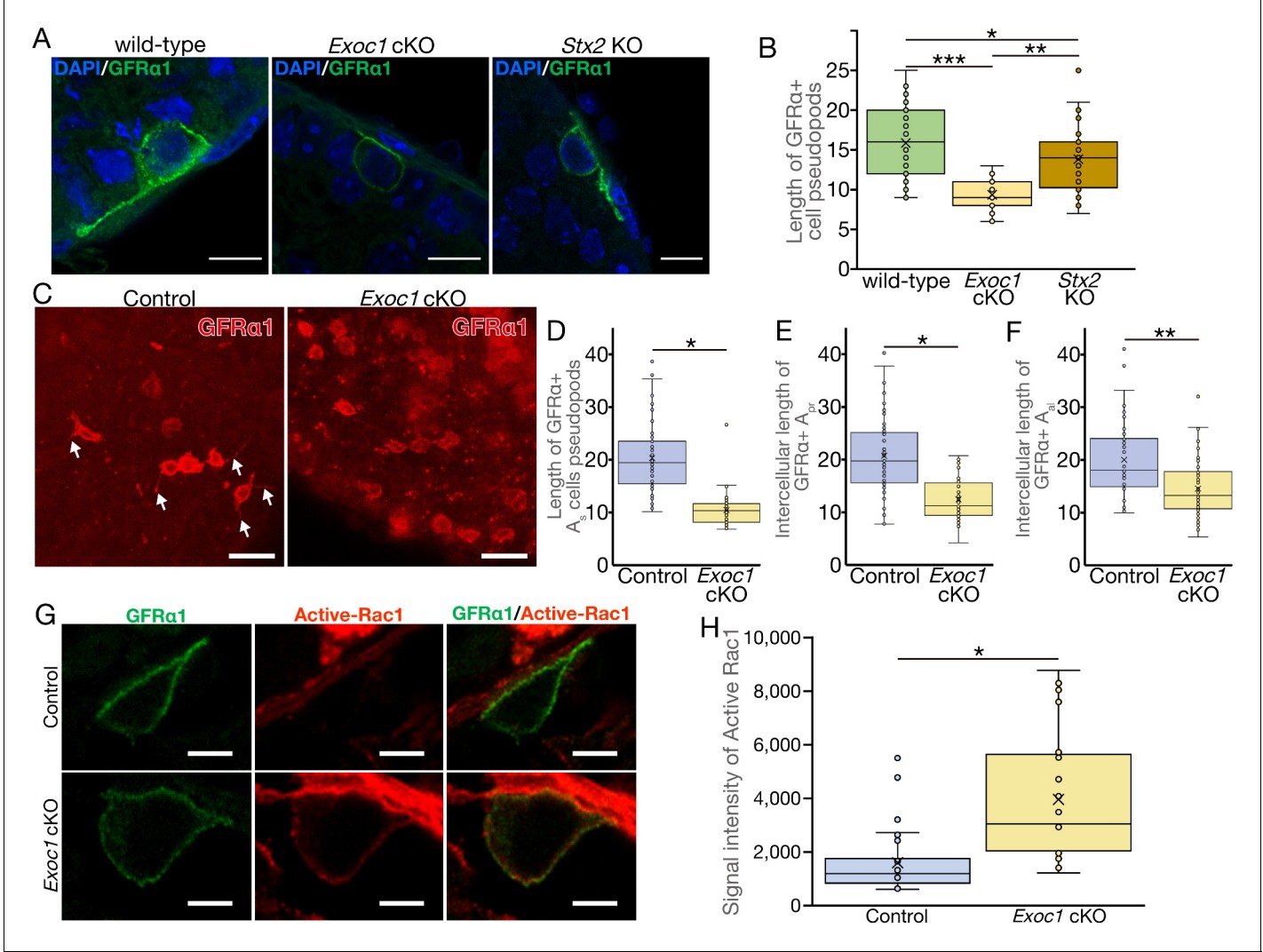

**Figure 4.** EXOC1 regulates pseudopod elongation via Rac1 inactivation. (**A**) A representative image of GFRα1⁺ undifferentiated spermatogonia in *Exoc1* cKO and *Stx2* KO. Pseudopod elongation was impaired in *Exoc1* cKO, but not in *Stx2* KO. Scale bars: 10 µm. (**B**) Pseudopod length quantification using sections. Average length of GFRα1⁺ spermatogonia pseudopods in *Exoc1* cKO was shorter than that of *Stx2* KO and wild type (n = 3 in each genotype, 25–36 cells in each mouse). *p=0.052, **p=1.8 × 10⁻⁶, ***p=9.5 × 10⁻⁹. one-way ANOVA. (**C**) Pseudopod length quantification through whole-mount immunofluorescence staining of adult testes. GFRα1⁺ spermatogonia with elongated pseudopod (white arrows) were frequently observed in control (*Exoc1^{flox/cKO}*) mice, whereas they were rarely observed in *Exoc1* cKO mice. Scale bars: 30 µm. (**D**) Measurement of the length of pseudopod of A_single GFRα1⁺ cells based on whole-mount immunofluorescence images (n = 3 in each genotype, 20 cells in each mouse). *p=7.2 × 10⁻¹⁷, Student's t-test. Control: *Exoc1^{flox/cKO}*. (**E, F**) Measurement of intercellular length in connected A_pair (n = 3 in each genotype, 20 intercellular distances in each mouse) or A_aligned (n = 3 in each genotype, 14–19 intercellular distances in each mouse) based on whole-mount immunofluorescence images. *p=3.6 × 10⁻¹², **p=0.00014, Student's t-test. Control: *Exoc1^{flox/cKO}*. (**G**) A representative image of active-Rac1 in GFRα1⁺ spermatogonia of *Exoc1* cKO adult testis. In control mice (*Exoc1^{flox/wt}:: Nanos3^{+/Cre}*), active-Rac1 signal was lower than the detection limit. Non-polar active-Rac1 signal was detected in *Exoc1* cKO adult testis. Scale bars: 5 µm. (**H**) Quantification of signal intensity of active-Rac1 in GFRα1⁺ spermatogonia based on immunostaining images. The average intensity in each cell is higher in the *Exoc1* cKO group than that in the control group (n = 3 in genotype, 8–10 cells in each mouse). *p=0.000036, Student's t-test. Control: *Exoc1^{flox/flox}*.

The online version of this article includes the following source data and figure supplement(s) for figure 4:

**Source data 1.** Measurement of the length of the pseudopodia of GFRα1+ cells in section observation.

**Source data 2.** Measurement of the length of the pseudopodia of GFRα1+ cells in whole-mount observation.

**Source data 3.** Intensity of active Rac1 signal in each cell.

**Figure supplement 1.** The production of the *Stx2* KO.

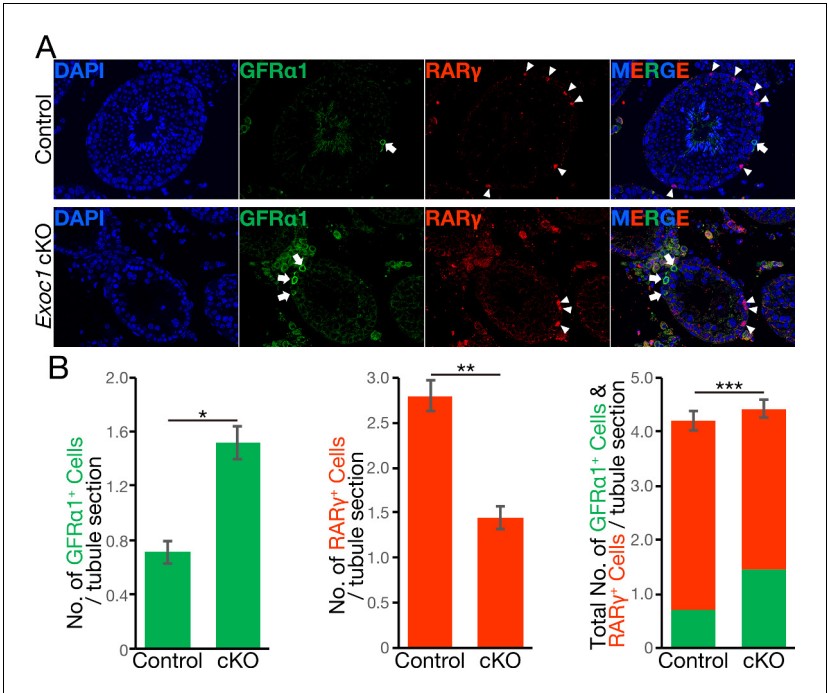

**Figure 5.** The balance of spermatogonial differentiation is perturbed in Exoc1 *cKO*. (**A**) Representative immunofluorescence image of adult *Exoc1* cKO seminiferous tubule. GFRα1⁺ spermatogonia (arrow) density in the cKO was higher than that in control mice. RARγ⁺ spermatogonia (arrowhead) density decreased in the cKO. Control: *Exoc1$^{flox/flox}$* mice. (**B**) The number of spermatogonia in a cross-section of seminiferous tubule (n = 3 in each genotype, 46–87 sections in each mouse). The number of GFRα1⁺ cell per section was significantly increased in the cKO. The number of RARγ⁺ cell in the cKO was significantly smaller than that in control (*Exoc1$^{flox/flox}$*) mice. *p=$1.9 \times 10^{-7}$, **p=$3.6 \times 10^{-9}$, ***p=0.035, Student's t-test.

The online version of this article includes the following source data and figure supplement(s) for figure 5:

**Source data 1.** The number of GFRα1+ and RARγ+ cells.

**Source data 2.** The number of c-Kit+ cells.

**Figure supplement 1.** The number of Kit⁺ differentiating spermatogonia are reduced in *Exoc1* cKO.

---

aggregation of spermatocytes. ICB complex proteins bind to small RNAs, suggesting that ICB may be involved in epigenetic regulation as well as the morphological formation of spermatocytes (*Iwamori et al., 2020*); in *Exoc1* cKO spermatocytes, this epigenetic regulation may be responsible for the arrest of spermatogenesis.

There are three possible reasons for spermatocyte aggregation due to impaired vesicle transport via EXOC1, STX2, and SNAP23. First, the impaired transport of sulfated glycolipids: 90% of the glycolipids in the testes are seminolipids (*Ishizuka, 1997*), and there are spermatocyte aggregates even in mice lacking the genes required for seminolipid biosynthesis (*Ugt8a* and *Gal3st1*) (*Fujimoto et al., 2000*; *Honke et al., 2002*). As the localization of seminolipids to the plasma membrane is impaired in spermatocytes of *Stx2*$^{repro34}$ mice (*Fujiwara et al., 2013*), similarly in *Exoc1* cKO mice, impaired seminolipid transport may have caused for the aggregation of the spermatocytes. Second, insufficient membrane addition: cytokinesis requires the expansion of the surface area of the cell with the formation of cleavage furrow. The exocyst localizes to ICB for local membrane addition (*Neto et al., 2013*), and mutant *Drosophila*, which lacks *Exoc8* function, shows little expansion of spermatocyte surface area causing the emergence of a defective cytokinetic ring in spermatocytes (*Giansanti et al., 2015*). This suggests that spermatocyte aggregation in *Exoc1* cKO mice is due to an impaired supply of cell membranes to facilitate cytokinesis. Third, the impaired transport of endosomal sorting complex required for transport (ESCRT) proteins that functions in cell separation: in cytokinesis, cells are separated by ESCRT proteins after the cleavage furrow is squeezed by a contractile ring (*Elia et al., 2011*). In cultured human cells, depletion of EXOC3 and EXOC4 impairs the transport of CHMP2B and CHMP4B, subunits of the ESCRT III complex, to ICB and causes

multinucleation (*Kumar et al., 2019*). This suggests that the exocyst complex is more important for recruitment of the ESCRT III complex to the ICB than for formation of ICB and that disruption of this recruitment in *Exoc1* cKO spermatocytes may be responsible for the impaired secondary ingression event (*Agromayor and Martin-Serrano, 2013*) in cytokinesis. On the Mouse Genome Informatics database (http://www.informatics.jax.org/), there are 10 genes encoding the ESCRT III complex sub-unit, and two of them (*Chmp2b* and *Chmp4c*) KO mice are fertile (*Dickinson et al., 2016*). The function of the other eight genes in spermatogenesis has not been analyzed (*Coulter et al., 2018*; *Lee et al., 2007*). Further research will be needed to test these possibilities.

Analysis of the differentiation status of spermatogonia showed an increase in GFRα1$^+$ spermatogonia and a decrease in RARγ$^+$ spermatogonia in *Exoc1* cKO mice, suggesting that spermatogonia are biased toward undifferentiated maintenance. In addition, the pseudopod of the GFRα1$^+$ spermatogonia was shorter in the *Exoc1* cKO mice and the fluorescence intensity of active-Rac1 was stronger. The exocyst is required for the transport of SH3BP1, which converts Rac1 from its active to inactive form, and over-activation of Rac1 inhibits cell migration (*Parrini et al., 2011*). Therefore, EXOC1 may regulate pseudopod elongation and migration in the undifferentiated spermatogonia by inactivating Rac1 via SH3BP1 transport; since there are no reports of *Sh3bp1* KO mice, it will be necessary to investigate whether pseudopod elongation and migration are impaired when *Sh3bp1* is deficient in undifferentiated spermatogonia. In addition, since undifferentiated spermatogonia move within the basal compartment of the seminiferous tubules and regulate the balance between self-renewal and differentiation by competing with each other for FGFs (*Kitadate et al., 2019*), undifferentiated spermatogonia migration may be important for maintaining the balance between self-renewal and differentiation. The disruption of this balance seen in *Exoc1* cKO mice may have been caused secondary to impaired pseudopod elongation of the cells. In other words, undifferentiated spermatogonia, which normally move within the basal compartment of the seminiferous tubules, replicate themselves near FGFs-producing cells, and differentiate into differentiation-primed spermatogonia by leaving FGFs-producing cells via cell migration. In contrast, *Exoc1* deficiency results in insufficient inactivation of Rac1 in undifferentiated spermatogonia and impaired its pseudopod elongation. As a result, undifferentiated spermatogonia that are restricted by migration may remain physically close to FGFs-producing cells and thus repeatedly self-renew, limiting their differentiation. It is still unclear whether all GFRα1$^+$ cells are included in the self-renewing spermatogonial stem cell compartment (*Lord and Oatley, 2017*). Moreover, at present, it is unclear whether Exoc1 contributes to the fate determination of all spermatogonial stem cells, and thus, further analysis is required.

# Materials and methods

## Key resources table

| Reagent type (species) or resource | Designation | Source or reference | Identifiers | Additional information |
|---|---|---|---|---|
| Strain, strain background (*M. musculus*) | C57BL/6J | Charles River Laboratories Japan | Stock No: 000664 | |
| Strain, strain background (*M. musculus*) | Crl:CD1(ICR) | Charles River Laboratories Japan | | |
| Genetic reagent (*M. musculus*) | *Exoc1 flox* | This paper | | This is from the *Exoc1$^{tm1a(EUCOMM)Hmgu}$*. IKMC Project #78575 |
| Genetic reagent (*M. musculus*) | *Exoc1$^{PA-C}$* | This paper | | |
| Genetic reagent (*M. musculus*) | *Exoc1$^{PA-N}$* | This paper | | |
| Genetic reagent (*M. musculus*) | *Snap23 flox* | This paper | | This is from the *Snap23$^{tm1a(EUCOMM)Wts}$*. Colony Name BLA3054 |

*Continued on next page*

*Continued*

| Reagent type (species) or resource | Designation | Source or reference | Identifiers | Additional information |
|---|---|---|---|---|
| Genetic reagent (*M. musculus*) | *Stx2 KO* | This paper | | |
| Genetic reagent (*M. musculus*) | *Nanos3$^{tm2\ (cre)Ysa}$* | **Suzuki et al., 2008** | Prof. Saga, RIKEN BRC (RDB13130) | RBRC02568 |
| Genetic reagent (*M. musculus*) | B6;SJL-Tg(ACTFLPe) 9205Dym/J | The Jackson Laboratory | Stock No: 003800 | |
| Cell line (Human) | HEK293T cell | ATCC | ATCC Sales Order: SO0623448 | FTA Barcode: STRB4056 |
| Antibody | Monoclonal anti-PA-tag, Biotin conjugated (Rat) | FUJIFILM Wako Chemicals | Cat#017–27731 | WB (1:500) |
| Antibody | Polyclonal anti-b-actin (Rabbit) | MEDICAL and BIOLOGICAL LABORATORIES | Cat#PM053 | WB (1:3000) |
| Antibody | Monoclonal anti-DYKDDDDK tag (Mouse) | FUJIFILM Wako Chemicals | Cat# 018–22381 | WB (1:2000) |
| Antibody | Monoclonal anti-DYKDDDDK tag (Rat) | FUJIFILM Wako Chemicals | Cat# 018–23621 | WB (1:2000) |
| Antibody | Monoclonal anti-HA-tag (Rabbit) | Cell Signaling Technology | Cat#3724 | WB (1:2000) |
| Antibody | Monoclonal anti-HA-tag (Mouse) | BioLegend | Cat#901513 | WB (1:1000) |
| Antibody | Monoclonal anti-Myc (Mouse) | MEDICAL and BIOLOGICAL LABORATORIES | Cat#M192-3 | WB (1:1000) |
| Antibody | Monoclonal anti-SNAP23 (Mouse) | Santa Cruz Biotechnology | Cat#sc-166244 | WB (1:1000) |
| Antibody | Anti-Rat IgG, HRP-linked (Goat) | GE Healthcare | Cat#NA935V | WB (1:30000) |
| Antibody | Anti-Rabbit IgG, HRP-linked (Donkey) | GE Healthcare | Cat#NA934V | WB (1:30000) |
| Antibody | Anti-Mouse IgG, HRP-linked (Sheep) | GE Healthcare | Cat#NA931V | WB (1:30000) |
| Antibody | Normal IgG (Rabbit) | FUJIFILM Wako Chemicals | Cat#148–09551 | Co-IP |
| Antibody | Polyclonal anti-SNAP23 (Rabbit) | Abcam | Cat#ab3340 | Co-IP |
| Antibody | Monoclonal anti-PA-tag (Rat) | FUJIFILM Wako Chemicals | Cat#016–25861 | IF (1:1000) |
| Antibody | Monoclonal anti-γH2AX (Mouse) | Merck-Millipore | Cat#05–636 | IF (1:100) |
| Antibody | Polyclonal anti-SYCP1 (Rabbit) | Novus Biological | Cat#NB300-299 | IF (1:50) |
| Antibody | Monoclonal anti-SYCP3 (Mouse) | Santa Cruz Biotechnology | Cat#sc-74569 | IF (1:50) |
| Antibody | Polyclonal anti-GFRα1 (Goat) | R and D systems | Cat#AF560 | IF (1:400 for section) IF (1:1000 for whole mount) |
| Antibody | Monoclonal anti-RARγ1 (Rabbit) | Cell Signaling Technology | Cat#8965S | IF (1:200) |
| Antibody | Monoclonal anti-active rac1 (Mouse) | NewEast Biosciences | Cat#26903 | IF (1:1000) |
| Antibody | Polyclonal anti-Exoc1 (Rabbit) | Proteintech | Cat#11690–1-AP | IF (1:50) |

*Continued*

| Reagent type (species) or resource | Designation | Source or reference | Identifiers | Additional information |
|---|---|---|---|---|
| Antibody | Polyclonal anti-Exoc1 (Rabbit) | Atlas Antibodies | Cat#HPA037706 | IF (1:50) |
| Antibody | Anti-Goat IgG, Alexa Fluor 488 (Chicken) | Thermo Fisher Scientific | Cat#A21467 | IF (1:200 for section) |
| Antibody | Anti-Goat IgG, Alexa Fluor 594 (Chicken) | Thermo Fisher Scientific | Cat#A21468 | IF (1:400 for whole mount) |
| Antibody | Anti-Rat IgG, Alexa Fluor 555 (Donkey) | Abcam | Cat#ab150154 | IF (1:1000) |
| Antibody | Anti-Mouse IgG, Alexa Fluor 555 (Goat) | Thermo Fisher Scientific | Cat#A28180 | IF (1:200) |
| Antibody | Anti-Mouse IgG, Alexa Fluor 555 (Donkey) | Thermo Fisher Scientific | Cat#A31570 | IF (1:200) |
| Antibody | Anti-Rabbit IgG, Alexa Fluor 647 (Goat) | Thermo Fisher Scientific | Cat#A27040 | IF (1:200) |
| Recombinant DNA reagent | pcDNA3.1 (+) Mammalian Expression Vector | Invitrogen | V79020 | |
| Recombinant DNA reagent | T7-NLS hCas9-pA plasmid | *Yoshimi et al., 2016* | RIKEN BRC (RDB13130) | |
| Recombinant DNA reagent | pCAG-Flpe | *Matsuda and Cepko, 2007* | Addgene (Plasmid #13787) | |
| Recombinant DNA reagent | pT7-Flpe-pA | This paper | RIKEN BRC (RDB16011) | |
| Sequence-based reagent | All primers in *Supplementary file 1b* | Thermo Fisher Scientific | | |
| Sequence-based reagent | All primers in *Supplementary file 1b* | Thermo Fisher Scientific | | |
| Sequence-based reagent | Exoc1 PA-C ssODN | Integrated DNA Technologies | | GAATTCACTATTCAGG ACATTCTGGATTATTGC TCCAGCATCGCACAG TCCCACGGCTCAACCAG CGGATCTGGTAAGCCAG GTAGTGGAGAAGGCAG CACCAAGCCTGGCGGCGT CGCCATGCCTGGAGCCGA GGATGATGTCGTGTAAGC CCTAGGAAAGAGGAGAAA GAAGTGAGCATGCATTCT CAGTCCAGCAAA |
| Sequence-based reagent | Exoc1 PA-N ssODN | Integrated DNA Technologies | | GGAGGGCAGTGGTTTTGAGAAT TATTCTAAATGTTTTTCAGCTG AGAAAAGATGGGCGTCGCCAT GCCTGGAGCCGAGGATGATGT CGTGGGCTCAACCAGCGGAT CTGGTAAGCCAGG TAGTGGAGAAGGC AGCACCAAGCCTGGC ACAGCAATCAAGCATGC GCTGCAGAGAGATATCTTC ACACCAAATGATGAACG |
| Software, algorithm | The R Foundation | https://www.r-project.org/foundation/ | | |
| Other | Streptavidin-HRP | Nichirei Biosciences | Cat#426061 | WB (1:1000 in 2% BSA/TBS-T) |
| Other | Lectin from Arachis hypogaea, FITC | Sigma-Adrich | Cat#L7381 | Lectin staining (1:100) |

## Animals

The mice were maintained in plastic cages under specific pathogen-free conditions in 23.5 ± 2.5℃ and 52.5 ± 12.5% relative humidity under a 14 hr light/10 hr dark cycle in the Laboratory Animal Resource Center at the University of Tsukuba. Mice had free access to commercial chow (MF diet; Oriental Yeast Co. Ltd) and filtered water. ICR and C57BL/6 mice were purchased from Charles River Laboratories Japan. *Nanos3-Cre* mice, *Nanos3^tm2 (cre)Ysa*, were kindly gifted by Dr. Saga (*Suzuki et al., 2008*) through RIKEN BioResource Research Center (RBRC02568). *Exoc1^tm1a (EUCOMM) Hmgu* and *Snap23^tm1a(EUCOMM)Hmgu* mice were obtained from the International Knockout Mouse Consortium and the International Mouse Phenotyping Consortium (*Skarnes et al., 2011*). All male mice used in the experiment were 10–20 weeks old. The genetic background of all the genetically modified or wild-type mice used in the experiments was C57BL/6, except for the *Nanos3*-Cre mice. Since the genetic background of the *Nanos3*-Cre mice provided by RIKEN BRC was ICR, they were backcrossed to C57BL/6 for at least five generations before being used in the experiment.

## Genome editing in mouse embryos

The *Exoc1* PA-C KI mice were generated by CRISPR-Cas9 based genome editing. The sequence (5′-GCA CAG TCC CAC TAA GCC CT-3′) was selected as the guide RNA (gRNA) target. The gRNA was synthesized and purified by the GeneArt Precision gRNA Synthesis Kit (Thermo Fisher Scientific, Waltham, Massachusetts) and dissolved in Opti-MEM (Thermo Fisher Scientific, Waltham, Massachusetts). The *Exoc1* PA-N KI mice were generated by CRISPR-Cas12a-based genome editing. The following sequence was selected as the crRNA target and synthesized artificially (Integrated DNA Technologies): 5′-AGC TGA GAA AAG ATG ACA GCA-3′. In addition, we designed a 200-nt single-stranded DNA oligonucleotide (ssODN) donor; the LG3 linker (*Kagoshima et al., 2007*) and the PA tag sequence was placed between 55-nt 5′- and 53-nt 3′-homology arms. This ssODN was ordered as Ultramer DNA oligos (Integrated DNA Technologies) and dissolved in Opti-MEM. Two gRNA targets (5′-CAT AAA GTG GTT GCG CTC TT-3′ and 5′-GCA GAT GTG ATG CTC GGC TG-3′) located on intron 4 and 5 of *Stx2*, respectively, were selected for producing the *Stx2* KO mouse. These two gRNAs were synthesized, purified, and dissolved in Opti-MEM as mentioned above.

The mixture of gRNA for *Exoc1* PA-C (25 ng/µL), ssODN (100 ng/µL), and GeneArt Platinum Cas9 Nuclease (Thermo Fisher Scientific, Waltham, Massachusetts) (100 ng/µL) or the mixture of crRNA for *Exoc1* PA-N (25 ng/µL), ssODN (100 ng/µL), and Alt-R A.s. Cas12a Nuclease V3 (Integrated DNA Technologies) (100 ng/µL) were electroplated to the zygotes of C57BL/6J mice (Charles River Laboratories Japan, Yokohama, Japan) by using the NEPA 21 electroplater (Nepa Gene Co. Ltd., Ichikawa, Japan) to produce the *Exoc1* PA KI mouse (*Sato et al., 2018*). The mixture of two gRNAs for *Stx2* (25 ng/µL, each) and GeneArt Platinum Cas9 Nuclease (100 ng/µL) were electroplated to zygotes for producing the *Stx2* KO mouse. After electroporation, two cell embryos were transferred into the oviducts of pseudopregnant ICR female and newborns were obtained.

## Electroporation of mice zygotes with *Flpe* mRNA

A *pT7-Flpe-pA* plasmid (deposited in RIKEN BioResource Research Center, RDB16011) was constructed from a *T7-NLS hCas9-pA* plasmid, which was kindly gifted by Dr. Mashimo (*Yoshimi et al., 2016*), through RIKEN BioResource Research Center (RDB13130). *Cas9* cDNA in the *T7-NLS hCas9-pA* was replaced with an *Flpe* cDNA from *pCAG-Flpe*. This *pCAG-Flpe* was kindly gifted by Dr. Cepko (*Matsuda and Cepko, 2007*) through Addgene (Plasmid #13787). *Flpe* mRNA was transcribed in vitro from *Nhe*I digested *pT7-Flpe-pA* by using mMESSAGE mMACHINE T7 ULTRA Transcription Kit (Thermo Fisher Scientific, Waltham, Massachusetts).

Female C57BL/6 mice (12 weeks old) were superovulated by intraperitoneal administration of 5 units of pregnant mare serum gonadotropin (ASKA Pharmaceutical Co. Ltd., Tokyo, Japan) and 5 units of human chorionic gonadotropin (ASKA Pharmaceutical Co. Ltd., Tokyo, Japan) with a 48 hr interval. In vitro fertilization was performed using the wild-type C57BL/6 oocytes and the *Snap23^tm1a (EUCOMM)Hmgu* sperms according to standard protocols. Five hours later, *Flpe* mRNA (300 ng/µL) was electroplated to the zygotes by using the NEPA 21 electroplater. The poring pulse was set to voltage: 225 V, pulse width: 2 ms, pulse interval: 50 ms, and number of pulses:+4. The transfer pulse was set to voltage: 20 V, pulse width: 50 ms, pulse interval: 50 ms, and number of pulses: ±5

(attenuation rate was set to 40%). A day after electroporation, the developed two-cell embryos were transferred to pseudopregnant ICR mice.

## Genotyping PCR

For routine genotyping, genomic DNA was extracted from <0.5 mm tails of 3 weeks old mice. PCR was carried out using AmpliTaq Gold 360 Master Mix (Thermo Fisher Scientific, Waltham, Massachusetts) with the appropriate primers (*Supplementary file 1b*). For genotyping of blastocysts, they were treated with a proteinase K (F. Hoffmann-La Roche, Basel, Switzerland) solution prepared in DDW (0.02 mg/mL), at 55°C for 2 hr and then at 95°C for 7 min. This solution was used as a template, to which AmpliTaq Gold 360 Master Mix and appropriate primers (*Supplementary file 1b*) were added for PCR. For DNA sequencing, PCR products were purified with a FastGene Gel/PCR Extraction Kit (Nippon Genetics, Tokyo, Japan) and sequences were confirmed with a BigDye Terminator v3.1 Cycle Sequencing Kit (Thermo Fisher Scientific, Waltham, Massachusetts), FastGene Dye Terminator Removal Kit (Nippon Genetics, Tokyo, Japan), and 3500xL Genetic Analyzer (Thermo Fisher Scientific, Waltham, Massachusetts).

## Western blot analysis and Co-IP of in vivo samples

Mouse testes were homogenized in tissue protein extraction reagent (Thermo Fisher Scientific, Waltham, Massachusetts) and centrifuged at 14,000 rpm for 15 min at 4°C. Protein concentrations of the supernatants were determined using Quick Start Bradford assay kit (BioRad). The samples were mixed with sample buffer solution containing a reducing reagent (6×) for SDS-PAGE (Nacalai Tesque Inc, Kyoto, Japan), followed by incubation for 3 min at 100°C. The samples were subjected to SDS-PAGE and transferred to Immobilon-P PVDF membranes (Merck-Millipore, Burlington, Massachusetts). The membranes were blocked with 5% skim milk in Tris-buffered saline/Tween 20 (TBS-T) for 30 min at 20–25°C. The membranes were subsequently incubated with primary antibodies (Key Resources Table) for 1–3 hr at 20–25°C. The membrane was then incubated with streptavidin-HRP or HRP-linked secondary antibodies (Key Resources Table) for 1 hr at 20–25°C. All antibodies were diluted in TBS-T with 1% skim milk. The blots were developed by chemiluminescence using Luminata Forte Western HRP Substrate (Merck-Millipore, Burlington, Massachusetts) or ImmunoStar LD (FUJIFILM Wako Chemicals, Tokyo, Japan) and visualized by iBrightCL100 (Thermo Fisher Scientific, Waltham, Massachusetts).

For Co-IP assay, proteins were extracted from testes using lysis buffer (20 mM Tris-HCl, 150 mM NaCl, 1 mM EDTA, 1% triton X-100, and protease inhibitor cocktail). The lysates containing 1.5 mg of total protein were incubated with normal rabbit IgG or anti-SNAP23 antibody (Key Resources Table) for 1 hr at 4°C. The samples were further incubated with SureBeads Protein G Magnetic Beads for 1 hr at 4°C. After washing the beads, proteins were eluted by boiling in 1× SDS sample buffer and subjected to SDS-PAGE.

## H and E staining

Testes, with the tunica albuginea removed, were fixed using 10%-Formaldehyde Neutral Buffer Solution (Nacalai Tesque Inc, Kyoto, Japan) overnight. Fixed testes were then soaked in 70% ethanol and embedded into paraffin blocks. Tissues were sliced at 5 μm using a HM335E microtome (Thermo Fisher Scientific, Waltham, Massachusetts). After deparaffinization, xylene was removed from the sections with 100% ethanol and subsequently hydrated with 95% ethanol, 70% ethanol, and deionized distilled water. Hydrated sections were stained with Mayer's Hematoxylin Solution (FUJIFILM Wako Chemicals, Tokyo, Japan) and 1% Eosin Y Solution (FUJIFILM Wako Chemicals, Tokyo, Japan).

## Electron microscope observation

Testes were fixed in 1% $OsO_4$ in 0.1 M phosphate buffer (pH 7.4), dehydrated in ethanol, and embedded in epoxy resin poly/Bed 812 (Polysciences Inc, Warrington, Pennsylvania). Subsequently, ultra-thin sections were made at 70–80 nm and stained with uranium acetate and lead citrate. A transmission electron microscope, JEM-1400 (JOEL Ltd., Tokyo, Japan), was used for the observation.

## Immunofluorescence and lectin staining

All immunofluorescence experiments except for active-Rac1, SYCP3, and SYCP1 were performed with paraffin sections. The paraffin sections were prepared similar to that of the H and E staining. After deparaffinization and rehydration, sections were permeabilized with 0.25% TritonX-100 in PBS and autoclaved (121°C, 10 min) with Target Retrieval Solution (Agilent Technologies, Santa Clara, California). Sections were incubated with Blocking One Histo (Nacalai Tesque Inc, Kyoto, Japan) for 15 min at 37°C. Primary antibodies (Key Resources Table) that were diluted with Can Get Signal Immunoreaction Enhancer Solution A (Toyobo Co. Ltd., Osaka, Japan) were applied and slides were incubated for 1 hr at 37°C. Alexa Fluor-conjugated secondary antibodies (Key Resources Table) were diluted with Can Get Signal Immunoreaction Enhancer Solution A, and applied for 1 hr at 37°C. Prolong gold antifade reagent with DAPI (Thermo Fisher Scientific, Waltham, Massachusetts) was used as a mounting media and for DAPI staining. Active-Rac1 immunofluorescence was performed with frozen sections. To prepare frozen sections, the tunica albuginea was removed from the testes, and the testes were fixed with 4% paraformaldehyde overnight at 4°C. Fixed testes were then soaked in 30% sucrose in PBS overnight at 4°C and embedded in Tissue-Tek O.C.T. Compound (Sakura Finetek, Tokyo, Japan). Tissues were sliced at 14 μm using HM525 NX Cryostat (Thermo Fisher Scientific, Waltham, Massachusetts). Permeabilization, blocking, and antibody reactions were performed similar to that used in immunofluorescence with paraffin sections. Whole-mount immunofluorescence of seminiferous tubules for GFRα1 (Key Resources Table) was performed as reported previously (*Kitadate et al., 2019*). Meiotic pachytene chromosome spread and immunofluorescence for SYCP1 and SYCP3 (Key Resources Table) was performed as reported previously (*Peters et al., 1997*). Samples were observed under a BZ-9000 fluorescence microscope (Keyence, Osaka, Japan), a SP8 Confocal Laser Scanning Microscopy (Leica microsystems, Wetzlar, Germany), and an LSM 800 with Airyscan (ZEISS, Oberkochen, Germany). PNA-lectin (Key Resources Table) staining was performed similar to that of immunofluorescence.

## Transfection and Co-IP

Testes RNA was extracted from C57BL/6 mouse using NucleoSpin RNA Plus (Nacalai Tesque Inc, Kyoto, Japan). cDNA was synthesized with PrimeScript RT Master Mix (Takara Bio, Kusatsu, Japan) and full-length cording sequences (CDS) of *Exoc1*, *Snap23*, and *Stx2* were obtained with PrimeSTAR GXL DNA Polymerase (Takara Bio, Kusatsu, Japan) using the appropriate primers (*Supplementary file 1b*). Full-length CDSs were introduced into the *pCDNA3.1* mammalian expression vector with epitope-tag sequences. Expression vectors for FLAG-EGFP, V5-mCherry-HA, and E2 crimson-MYC-His were used as negative controls. These vectors were transfected into HEK293T cells with PEI MAX (Polysciences Inc, Warrington, Pennsylvania) in 100 mm culture dish according to the manufacturer's protocol. After 24 hr incubation, cells were lysed in lysis buffer (20 mM Tris-HCl, 150 mM NaCl, 1 mM EDTA, 1% triton X-100, protease inhibitor cocktail). The cell lysates were centrifuged at 14,000 rpm for 10 min at 4°C. The supernatants were used for immunoprecipitation. Co-IP was performed with a HA-tagged protein purification kit, DDDDK-tagged protein purification kit, and c-Myc-tagged protein mild purification kit ver.2 (Medical and Biological Laboratories Co., Nagoya, Japan) according to the manufacturer's protocol. The antibodies for western blot are listed in Key Resources Table. HEK293T cells were obtained from The American Type Culture Collection (Manassas, Virginia). The HEK293T cell line has been authenticated by STR profiling and is negative of mycoplasma contamination testing.

## Quantification of pseudopod elongation with thick sections

Immunofluorescence for GFRα1 was performed using thinly sectioned (20 μm) paraffin sections. To include the entire cell, Z-stack photography was performed using a SP8 Confocal Laser Scanning Microscopy (Leica microsystems, Wetzlar, Germany). We applied Sholl analysis (*Binley et al., 2014*) to quantify pseudopod elongation, drawing concentric circles around the midpoint of cell body width and the intersection with the basement membrane and measuring the distance of the furthest elongated pseudopod.

## Image analyses

The elongation of the pseudopod and the distance between connected cells in whole-mount immunofluorescence assay, signal intensity of active-Rac1immunofluorescence, and the area of seminiferous tubule sections were calculated using ImageJ Fiji 1.53 c JAVA 1.8.0_172 (64-bit). All image data were selected randomly. For pseudopod extension analysis, the distance between the center ('Centroid') of the cell body and the tip of the pseudopod, at the furthest point from the cell body, was measured. For the analysis of the distance between connected cells, the distance between the centers ('Centroid') of each cell body was measured. For intensity analysis, we measured the intensity of active-Rac1 in GFR$\alpha$1$^+$ cells using immunofluorescence images. After selecting GFR$\alpha$1$^+$ cells, we calculated the signal intensity of active-Rac1, which was converted from RGB to 16-bit. The average intensity of the signal ('Mean Gray Value') was calculated so that the size of the cells would not affect the measurement.

## Analysis of publicly available next-generation sequencing data

The expression of each exocyst subunit in the respective germline differentiation stage was extracted from the open data source GSE112393 (*Green et al., 2018*). The expression of each exocyst subunit in Sertoli cells of adult mice was extracted from GSM3069461 (*Green et al., 2018*). The gene count was log-normalized and visualized in comparison with the expression levels of all genes. The R script used in the analysis of GSM3069461 can be downloaded from https://github.com/aki-kuno/Exoc_GSM3069461/blob/master/SupFig2.R.

## Study approval

All animal experiments were carried out in a humane manner with approval from the Institutional Animal Experiment Committee of the University of Tsukuba in accordance with the Regulations for Animal Experiments of the University of Tsukuba and Fundamental Guidelines for Proper Conduct of Animal Experiments and Related Activities in Academic Research Institutions under the jurisdiction of the Ministry of Education, Culture, Sports, Science, and Technology of Japan (Approval Number: 20–013).

## Acknowledgements

We would like to thank Tokuko Iwamori for advice on the experimental design for syncytia analyses. We are grateful to Narumi Ogonuki and Atsuo Ogura for their helpful discussions. We would like to thank Yoshihiro Miwa, Hiroyuki Sakuma, and Akio Sekikawa for advice on the experimental design for fluorescence imaging. We are grateful to Aya Ikkyu and Tomoyuki Fujiyama for advice on the experimental design for protein analyses. This work was supported by Scientific Research (B) (17H03566: KY and 19H03142: SM) from the Ministry of Education, Culture, Sports, Science, and Technology (MEXT).

## Additional information

### Funding

| Funder | Grant reference number | Author |
| --- | --- | --- |
| Ministry of Education, Culture, Sports, Science and Technology | 17H03566 | Ken-ichi Yagami |
| Ministry of Education, Culture, Sports, Science and Technology | 19H03142 | Seiya Mizuno |

The funders had no role in study design, data collection and interpretation, or the decision to submit the work for publication.

## Author contributions
Yuki Osawa, Investigation, Writing - original draft; Kazuya Murata, Investigation, Methodology; Miho Usui, Yumeno Kuba, Hoai Thu Le, Natsuki Mikami, Yoshihisa Ikeda, Kento Morimoto, Investigation; Toshinori Nakagawa, Shosei Yoshida, Supervision, Methodology; Yoko Daitoku, Kanako Kato, Yoko Tanimoto, Resources, Methodology; Hossam Hassan Shawki, Masatsugu Ema, Satoru Takahashi, Resources, Supervision; Akihiro Kuno, Methodology; Tra Thi Huong Dinh, Resources; Ken-ichi Yagami, Resources, Supervision, Funding acquisition; Seiya Mizuno, Conceptualization, Funding acquisition, Writing - original draft, Project administration; Fumihiro Sugiyama, Supervision, Funding acquisition

## Author ORCIDs
Tra Thi Huong Dinh (iD) http://orcid.org/0000-0003-1705-3865
Shosei Yoshida (iD) http://orcid.org/0000-0001-8861-1866
Seiya Mizuno (iD) https://orcid.org/0000-0002-6740-5817
Fumihiro Sugiyama (iD) http://orcid.org/0000-0003-4744-3493

## Ethics
Animal experimentation: All animal experiments were carried out in a humane manner with approval from the Institutional Animal Experiment Committee of the University of Tsukuba in accordance with the Regulations for Animal Experiments of the University of Tsukuba and Fundamental Guidelines for Proper Conduct of Animal Experiments and Related Activities in Academic Research Institutions under the jurisdiction of the Ministry of Education, Culture, Sports, Science, and Technology of Japan.

## Decision letter and Author response
Decision letter https://doi.org/10.7554/eLife.59759.sa1
Author response https://doi.org/10.7554/eLife.59759.sa2

# Additional files

### Supplementary files
• Supplementary file 1. Data of differentiating spermatogonia aggregation and primers for genotyping. (a) Differentiating spermatogonia are not aggregated. Examination of aggregated Kit$^+$ differentiating spermatogonia in adult *Exoc1* cKO mice. Aggregation was determined by observation of immunofluorescence images with anti-Kit antibody. (b) Primers for genotyping The following is a list of primers used for genotyping or vector construction.

• Transparent reporting form

### Data availability
All data generated or analysed during this study are included in the manuscript and supporting files.

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
