## [Decision Letter]

**Acceptance summary:**

Male germ cells undergo profound morphological changes during spermatogenesis, and highly orchestrated pieces of machinery are required for this process. These morphological changes require a formation of syncytia, and the failure of forming it results in the arrested spermatogenesis and causes male infertility. The authors have conducted a series of well-designed and labor-intensive experiments using genome-edited mouse lines and provided evidence that EXOC1 gene, which encodes a member of the Exocyst complex, is essential for the morphological changes of male germ cells during spermatogenesis. They showed that EXOC1 inactivates the Rho family small GTPase Rac1, and also plays a role in syncytia formation acting together with SNARE proteins.

**Decision letter after peer review:**

Thank you for submitting your article "EXOC1 regulates cell morphology of spermatogonia and spermatocytes in mice" for consideration by *eLife*. Your article has been reviewed by 3 peer reviewers, and the evaluation has been overseen by a Reviewing Editor and Anna Akhmanova as the Senior Editor. The following individuals involved in review of your submission have agreed to reveal their identity: Robin Hobbs (Reviewer #2); Stefan Schlatt (Reviewer #3).

The reviewers have discussed the reviews with one another and the Reviewing Editor has drafted this decision to help you prepare a revised submission.

Summary:

The study by Osawa and colleagues highlights interesting and highly desired aspects of spermatogonial physiology and clonal expansion. Specifically, the role of EXOC1, a factor involved in syncytial organization of cells, is explored and many novel aspects are described. The authors have conducted a series of well-designed and labor-intensive experiments using genome-edited mouse lines, and successfully evidenced that germ cell EXOC1 in the testis is essential for their morphological changes during spermatogenesis therein. This manuscript is focused on an understudied and critical pathway in the male germline and would be of interest to the field and of relevance for fertility. However, in its current state, the insight provided by the manuscript is somewhat limited. The authors need to perform a much more detailed characterization of the mouse models generated and revisit the model proposed for exocyst function in the germline. The manuscript is well written and clear but short of mechanistic points. In contrast to the strong genetic studies, however, the referees see some weakness in their biochemical, histological, and physiological studies as indicated below.

Essential revisions:

1. The macroscopic images for HE stained testicular sections should be included otherwise it is difficult to judge the effects of genetic ablation. And the authors should present how efficiently genes are inactivated in these mice.

2. Line 134 (Figure 1C): Authors are recommended to distinguish germ- and somatic-cells in testis.

3. The authors only showed representative images. Quantitative data need to be included.

4. Figure 3A is the only data showing EXOC1, STX2, and SNAP23. The authors should include negative control such as other proteins tagged with HA. More importantly, the authors are strongly recommended to show the interaction in vivo (IP with HA then detect by IB or MS analysis).

5. Line 301: The authors suggest three possible reasons to explain the phenotype. As shown in the previous study, the author can easily address if lipids accumulate in mutant cells.

6. Expression pattern of Exoc1 in the germline requires additional analysis. Expression of the PA tagged EXOC1 from knock in mice should be compared to known markers of germ cell and somatic cell populations. Expression in these populations also requires quantification and comparison to seminiferous epithelium stages. Previously published testis single cell RNA-Seq datasets should be analyzed to assess expression of Exoc1 and other exocyst complex proteins during spermatogenesis and in supporting somatic cells. The western blot of Figure 1B requires additional replicates and molecular weights of bands indicated.

7. Multiple conditional knockout mouse models have been developed for the manuscript. However, the efficiency of gene knockout in germline cells needs to be confirmed for each model. Inefficient gene knockout can affect interpretation of phenotype. For example, the mild phenotype shown by Snap23 knockout model.

8. The PA tagged EXOC1 model is a novel line generated by CRISPR-based methods for the manuscript. Additional evidence needs to be provided that the genomic sequence is correctly modified and tag sequence in frame etc. Further, why is the LG3 linker region included?

9. Phenotype of the conditional knockout Exoc1 and Snap23 models are intriguing. However, no quantification is provided for the analyses. How frequently are the multinucleate meiotic spermatocytes observed? What ages of animals are studied? Both prepubertal and adult mice need to be analyzed to account for defects in postnatal germline development. Distinct germ cell populations need to be quantified by use of known markers (spermatogonia, spermatocytes, spermatids). How many KIT+ cells were analyzed to check for cell aggregation? How many chromosome spreads of spermatocytes were analyzed for the Exoc1 knockout? Representative images of knockout and control spreads should be provided in the figure.

10. Based on EM analysis of multinucleate spermatocytes from Exoc1 knockout testis (please provide quantification for this result), the authors suggest that intercellular bridge (ICB) formation in spermatocytes is specifically disrupted. This proposed model for exocyst complex function a little confusing. For formation of an ICB, cytokinesis is initiated at the end of mitosis but recruitment of TEX14 to the midbody blocks abscission and mid-body components remain as components of the ICB. The exocyst complex has a known role in cytokinesis, suggesting that cytokinesis is not initiated in the knockout and multinucleate cells are therefore formed. ICBs are then of course not present but this reflects the role of EXOC1 in the cytokinesis pathway rather than ICB formation per se. The authors should clarify this model.

11. For IP analysis in figure 3, the reciprocal pull-downs need to be performed. For instance, IP anti-FLAG and anti-Myc tag and test for interactions. Molecular weights for proteins need to be included.

12. Roles for EXOC1 in pseudopod formation in GFRa1-positive SSCs are suggested (Figure 4). While very interesting, it seems that correctly identifying pseudopods based on GFRa1 membrane staining of sections might be challenging. Particularly as GFRa1 cells can be present as interconnected pairs of cells (and some 4 cell chains), which can disrupt the ability to identify pseudopods of individual cells. The authors should repeat this analysis using wholemount staining of seminiferous tubules and compare the distinct GFRa1-positive cell populations (single, pairs and aligned). This will allow better quantification of cell extensions. Can specific markers of pseudopods be used (even just actin staining etc)? The active-Rac1 staining in panel D also needs to be quantified. Wholemount analysis can also reveal whether EXOC1 plays a role in cytokinesis of spermatogonial populations.

13. Defects in migration of Exoc1 knockout GFRa1-positive SSCs are suggested to result in SSC accumulation and defects in production of differentiation-primed undifferentiated spermatogonia (Figure 5). Images shown in panel A indicate that clumps of GFRa1-positive cells accumulate, with some cells pushed off the basal layer of the seminiferous tubules. This is highly unusual as all spermatogonia should be adherent to the basement membrane. It seems unlikely that this phenotype is due solely to defects in migration. Additional data should be provided to support this model and exclude other potential defects. For example, can functional data (e.g. in vitro analysis) be provided to confirm issues with SSC migration? Also, as fewer differentiation-primed cells are generated, the numbers of differentiating spermatogonia should be reduced in the knockout. Is this the case?

14. The manuscript text should be checked carefully for accuracy throughout. For instance, in the introduction, second paragraph, spermatogonia are stated to be divided into undifferentiated, differentiation-primed and differentiated fractions. However, differentiation -primed cells are found within the undifferentiated population as are SSCs. The significance of markers used for different spermatogonial populations (e.g. GFRa1) should be included in the introduction for clarity. The discussion should be modified to account for new datasets and models.

15. Many datasets are not quantified (see specific points above). Statistics cannot be evaluated as for most experimental settings the basic information (N number, means and coefficient of Variation… ) is not provided.

16. Additionally, the actual study design is not described. Therefore, the validity of the findings is partially questionable. No Information is provided on the number of mice analyzed. Nature and size of experimental groups and controls are not mentioned. Strategies for histological analysis are poorly described. Did the authors use random systematic sampling approaches on sufficient number of samples? The excellent micrographs must be considered individual observations as generalization of the descriptions would only be valid if repetitions are reported. Similar concerns apply to the blotting results. To check for validity and statistically Sound analysis, the authors need to provide a flow chart or table of all experimental animals and the endpoints analyzed in specific groups.

17. The mouse models are not as well defined as the authors Claim. Snap23 k/O mice Show germ cell development up to spermatids (Figure 3). Can the authors elaborate more on the presence of multinucleated cells and spermatids at the same time? Was that stable in all animals of this genotype?

[Editors' note: further revisions were suggested prior to acceptance, as described below.]

Thank you for resubmitting your work entitled "EXOC1 plays an integral role in spermatogonia pseudopod elongation and spermatocyte stable syncytium formation in mice" for further consideration by *eLife*. Your revised article has been reviewed by three peer reviewers and the evaluation has been overseen by a Reviewing Editor and Anna Akhmanova as the Senior Editor.

The manuscript has been significantly improved but there are some remaining issues (many could be amended with textual revisions) that need to be addressed, as outlined below:

1. Fluorescent images are still not clear.

2. Line 161: The authors mentioned "EXOC1 is observed in every cell in the adult testes" in the Figure 1 legend but described "EXOC1-PA was also detected in all male germ cells observed" in text. They may want to focus on germ cells, but it's better clarifying both Sertoli and germ cells are positive in the text if all cells are stained in testicular tubules.

The authors could not judge if Sertoli cells are positive with the current figure.

3. It was originally requested for the authors to assess the efficiency of floxed gene knockout in their models. While they have assessed whether remaining sperm in the Exoc1 conditional KO model are gene deleted, analysis of testis from this and other KO models described has not been performed. A simple qRT-PCR on total testis extracts or isolated testis cells could be used. It is important to define efficiency of gene knockout as the phenotype of some of the models is mild, e.g. Snap23 KO.

4. A new N-terminally PA tagged Exoc1 model is described in the revision as the C-tagged version clearly disrupts EXOC1 function, which is embryonic lethal. Confusingly, while the homozygous N-tagged mice are viable, indicating that EXOC1 function is retained, the authors could not detect expression of the N-tagged EXOC1 by immunostaining and suggest that it might be unstable. This seems counterintuitive and text should be modified to help resolve and discuss this issue. On a related point, lines 160-163 where rationale for use of the LG3 tag in the fusion construct is described – this should be reworded for clarity.

5. It is stated that while GFRa1+ spermatogonial pseudopods in the Exoc1 KO are shorter, those of the Stx2 KO are not, suggesting that EXOC1 operates independently of STX2 in this context. However, the pseudopods of Stx2 KO are strongly trending to be shorter and P value is quoted as 0.052. This suggests that a larger dataset may indeed show significant reduction and the conclusions from this data are overstated. Text should be changed appropriately to reflect this strong trend in data and/or more cells quantified to assess the robustness of the conclusions.

6. The abstract could be reworded for clarity. A sentence about role of EXOC1, SNAP23 etc in the exocyst complex should be included up-front to help non experts in the field. The term "high regulation" in the first sentence is ambiguous.

*Reviewer #1:*

The authors addressed most of the reviewer's concern, but there remains some remaining issues.

1. Fluorescent images are still not clear. The reviewer doesn't see the improvement.

2. Line 161: The authors mentioned "EXOC1 is observed in every cell in the adult testes" in the Figure 1 legend but described "EXOC1-PA was also detected in all male germ cells observed" in text. They may want to focus on germ cells, but it's better clarifying both Sertoli and germ cells are positive in the text if all cells are stained in testicular tubules.

The authors could not judge if Sertoli cells are positive with the current figure.

*Reviewer #2:*

The authors have performed substantial revisions to the manuscript, which is now significantly improved. In particular, they have included quantification of multiple key datasets as requested and included new analyses of exocyst complex gene expression patterns, spermatogonia cytoplasmic extensions, plus more thorough in vitro datasets. In general, they have responded very well to the comments.

The manuscript contains a wealth of data from genetic models although still lacks substantial mechanistic insight. However, it would be of interest to the field.

1. It was originally requested for the authors to assess the efficiency of floxed gene knockout in their models. While they have assessed whether remaining sperm in the Exoc1 conditional KO model are gene deleted, analysis of testis from this and other KO models described has not been performed. A simple qRT-PCR on total testis extracts or isolated testis cells could be used. It is important to define efficiency of gene knockout as the phenotype of some of the models is mild, e.g. Snap23 KO.

2. A new N-terminally PA tagged Exoc1 model is described in the revision as the C-tagged version clearly disrupts EXOC1 function, which is embryonic lethal. Confusingly, while the homozygous N-tagged mice are viable, indicating that EXOC1 function is retained, the authors could not detect expression of the N-tagged EXOC1 by immunostaining and suggest that it might be unstable. This seems counterintuitive and text should be modified to help resolve and discuss this issue. On a related point, lines 160-163 where rationale for use of the LG3 tag in the fusion construct is described – this should be reworded for clarity.

3. It is stated that while GFRa1+ spermatogonial pseudopods in the Exoc1 KO are shorter, those of the Stx2 KO are not, suggesting that EXOC1 operates independently of STX2 in this context. However, the pseudopods of Stx2 KO are strongly trending to be shorter and P value is quoted as 0.052. This suggests that a larger dataset may indeed show significant reduction and the conclusions from this data are overstated. Text should be changed appropriately to reflect this strong trend in data and/or more cells quantified to assess the robustness of the conclusions.

4. The numbers of cells scored in some of the datasets e.g. levels of activated Rac1, are very small – 8-10 cells per mouse. I would recommend scoring substantially more cells per mouse to improve robustness of the data.

5. The abstract could be reworded for clarity. A sentence about role of EXOC1, SNAP23 etc in the exocyst complex should be included up-front to help non experts in the field. The term "high regulation" in the first sentence is ambiguous.

*Reviewer #3:*

This is a valuable insight into spermatogonial biology. The mouse models provide fundamental evidence for the involvement if specific factors in germ cell eplitting and expansion. The findings are relevant for stem cell and infertility Research.

The authors provide relevant new mouse K/O models showing the involvement of EXOC 1 in spermatogonial pseudopod formation. This is an important observation. The authors have undertaken a solid revision of the paper. While some aspects of the histological analysis remain weak, the manuscript contains tremendous datasets of proven validity. The information will lead to a better understanding of spermatogonial physiology and initiation of spermatogenesis.

---

## [Author Response]

Essential revisions:1. The macroscopic images for HE stained testicular sections should be included otherwise it is difficult to judge the effects of genetic ablation. And the authors should present how efficiently genes are inactivated in these mice.

Thank you for your comments, and we agree with your suggestion. We have added the macroscopic image data to Figure 2C and Figure 3C. We also added Figure 2—figure supplement 1 and Figure 3—figure supplement 2 as data on the frequency of AGS and the gene ablation efficiency. We described the contents of these data in Lines 183-195 and Lines 256-262 in the main text.

2. Line 134 (Figure 1C): Authors are recommended to distinguish germ- and somatic-cells in testis.

We have added data to Figure 1—figure supplement 1C showing that it is expressed in germ cells. We have also added data on expression in Sertoli cells as Figure 1—figure supplement 2. We described the contents of these data in Lines 160-162 in the main text.

3. The authors only showed representative images. Quantitative data need to be included.

Thank you for your suggestion. Quantitative data for each experiment is added to Supplementary File1a, Figure 4I and Figure 2—figure supplement 1, Figure 3—figure supplement 2, and Figure 5—figure supplement 1.

Supplementary File1a: Number of c-Kit^+^ AGS in *Exoc1* cKO

Figure 4I: Intensity of active Rac1

Figure 2—figure supplement 1: Frequency of AGS in *Exoc1* cKO

Figure 3—figure supplement 2: Frequency of AGS in *Snap23* cKO

Figure 5—figure supplement 1: The number c-Kit^+^ cells in *Exoc1* cKO

4. Figure 3A is the only data showing EXOC1, STX2, and SNAP23. The authors should include negative control such as other proteins tagged with HA. More importantly, the authors are strongly recommended to show the interaction in vivo (IP with HA then detect by IB or MS analysis).

The in vitro Co-IP results for all combinations, including the negative control, are displayed as Figure 3A. We described the contents of these data in Lines 235-237 in the main text.

The in vivo co-IP data is presented in Figure 3B, and we also performed an experiment to detect SNAP23 by IP with PA, but we could not detect it, probably due to the low sensitivity. We described the contents of these data in Lines 237-241 in the main text.

We provided data showing that PA-tagged EXOC1 is functional in Lines 137-155 of the main text. We found that the C-terminal PA-tagged EXOC1 was not functional, so we generated a new N-terminal tagged mouse (Figure 1A).

5. Line 301: The authors suggest three possible reasons to explain the phenotype. As shown in the previous study, the author can easily address if lipids accumulate in mutant cells.

DI8 antibodies to detect target lipids are not commercially available, so we have not conducted this study due to difficulty in availability. The claim of this study is that EXOC1 is involved in the STX2 pathway, and studies on the downstream of this pathway may be a future consideration.

6. Expression pattern of Exoc1 in the germline requires additional analysis. Expression of the PA tagged EXOC1 from knock in mice should be compared to known markers of germ cell and somatic cell populations.

We found almost same comment in the Essential revisions comment 2. We answered there.

Expression in these populations also requires quantification and comparison to seminiferous epithelium stages. Previously published testis single cell RNA-Seq datasets should be analyzed to assess expression of Exoc1 and other exocyst complex proteins during spermatogenesis and in supporting somatic cells. The western blot of Figure 1B requires additional replicates and molecular weights of bands indicated.

We confirmed that all Exocyst subunits were expressed in both Germ and Sertoli cells. These data are shown in Figure 1—figure supplement 1A and 2.

7. Multiple conditional knockout mouse models have been developed for the manuscript. However, the efficiency of gene knockout in germline cells needs to be confirmed for each model. Inefficient gene knockout can affect interpretation of phenotype. For example, the mild phenotype shown by Snap23 knockout model.

Thank you for your suggestion. We have confirmed that spermatogonia lacking *Exoc1* do not produce fertilizable sperm. The data was shown in Figure 2—figure supplement 1D. We described the contents of this data in Lines 189-195 in the main text. Since the distance of loxP in *Snap23* is almost the same as that of *Exoc1* (*Exoc1* 902 bp/*Snap23* 754 bp), it is considered that the *Snap23* gene function is disrupted with the same efficiency as *Exoc1*.

8. The PA tagged EXOC1 model is a novel line generated by CRISPR-based methods for the manuscript. Additional evidence needs to be provided that the genomic sequence is correctly modified and tag sequence in frame etc.

We present the Sanger sequence data on Figure 1—figure supplement 1B.

Further, why is the LG3 linker region included?

This is to increase flexibility. The reason for this is described in Lines 142-145 in the main text.

9. Phenotype of the conditional knockout Exoc1 and Snap23 models are intriguing. However, no quantification is provided for the analyses. How frequently are the multinucleate meiotic spermatocytes observed?

Thank you for your suggestion.

We found almost same comment in the Essential revisions comment 2. We answered there.

What ages of animals are studied? Both prepubertal and adult mice need to be analyzed to account for defects in postnatal germline development.

We analyzed adult mice. We have stated this in the description of each experiment.

We have confirmed that AGS also appears in the first wave. Please see Author response image 1. Since the studies on the appearance of AGS due to the loss of STX2 and the involvement of migration in the differentiation state of sperm stem cells that were the basis of this study mainly used adults, we decided to present only adult data.

Distinct germ cell populations need to be quantified by use of known markers (spermatogonia, spermatocytes, spermatids).

The frequencies of AGS-spermatocytes and Sperm are provided in Figure 2—figure supplement 1 and Figure 3—figure supplement 2. The frequencies of spermatogonia are shown in Figure 5 and Figure 5—figure supplement 1.

How many KIT+ cells were analyzed to check for cell aggregation?

This information is described in Supplementary File1a.

How many chromosome spreads of spermatocytes were analyzed for the Exoc1 knockout? Representative images of knockout and control spreads should be provided in the figure.

I have documented this information in Figure 2—figure supplement 2 and its figure legend.

10. Based on EM analysis of multinucleate spermatocytes from Exoc1 knockout testis (please provide quantification for this result).

Thank you for your suggestion.

We found almost same comment in the Essential revisions comment 2. We answered there.

The authors suggest that intercellular bridge (ICB) formation in spermatocytes is specifically disrupted. This proposed model for exocyst complex function a little confusing. For formation of an ICB, cytokinesis is initiated at the end of mitosis but recruitment of TEX14 to the midbody blocks abscission and mid-body components remain as components of the ICB. The exocyst complex has a known role in cytokinesis, suggesting that cytokinesis is not initiated in the knockout and multinucleate cells are therefore formed. ICBs are then of course not present but this reflects the role of EXOC1 in the cytokinesis pathway rather than ICB formation per se. The authors should clarify this model.

I appreciate your valuable suggestion. The following text is added to Lines 378-382 in the main text. “exocyst complex is more important for recruitment of the ESCRT III complex to the ICB than for formation of ICB and that disruption of this recruitment in *Exoc1* cKO spermatocytes may be responsible for the impaired secondary ingression event (Agromayor and Martin-Serrano, 2013) in cytokinesis.”

11. For IP analysis in figure 3, the reciprocal pull-downs need to be performed. For instance, IP anti-FLAG and anti-Myc tag and test for interactions. Molecular weights for proteins need to be included.

Thank you for your suggestion. We found almost same comment in the Essential revisions comment 4. We answered there.

12. Roles for EXOC1 in pseudopod formation in GFRa1-positive SSCs are suggested (Figure 4). While very interesting, it seems that correctly identifying pseudopods based on GFRa1 membrane staining of sections might be challenging. Particularly as GFRa1 cells can be present as interconnected pairs of cells (and some 4 cell chains), which can disrupt the ability to identify pseudopods of individual cells. The authors should repeat this analysis using wholemount staining of seminiferous tubules and compare the distinct GFRa1-positive cell populations (single, pairs and aligned). This will allow better quantification of cell extensions. Can specific markers of pseudopods be used (even just actin staining etc)?

I have provided that information in Figure 4 D-G and described it in the main text, Lines 294-305. These are also GFRα1 immunostaining. We also performed WGA and actin immunostaining to get a clearer picture of the cell shape, but all the cells were stained and we could not measure the length of the pseudopodia in a single cell.

The active-Rac1 staining in panel D also needs to be quantified. Wholemount analysis can also reveal whether EXOC1 plays a role in cytokinesis of spermatogonial populations.

Thank you for your suggestion. The quantified signal intensity of active Rac1 is displayed as Figure 4I.

13. Defects in migration of Exoc1 knockout GFRa1-positive SSCs are suggested to result in SSC accumulation and defects in production of differentiation-primed undifferentiated spermatogonia (Figure 5). Images shown in panel A indicate that clumps of GFRa1-positive cells accumulate, with some cells pushed off the basal layer of the seminiferous tubules. This is highly unusual as all spermatogonia should be adherent to the basement membrane. It seems unlikely that this phenotype is due solely to defects in migration. Additional data should be provided to support this model and exclude other potential defects. For example, can functional data (e.g. in vitro analysis) be provided to confirm issues with SSC migration?

I would like to thank you for your valued remarks. We measured the occurrence of such a multilayered cell population, but it was very low. Please see Author response image 2. Since such multiple layers are rare, we replaced the *Exoc1* cKO figure used in Figure 5A. In Author response image 2, we present all the cases (just only two) that we found. The top panel in Author response image 2 is the figures used in the Figure 5 of the first version. Considering the possibility that GFRα1^+^ spermatogonia in *Exoc1* cKO mice are in a more undifferentiated state than those in control mice, we compared the expression level of GFRα1 between the *Exoc1* cKO and control (*Exoc1^flox/flox^*) groups, but there was no noticeable difference (Author response image 2). In addition, although it was very rare case, Kit^+^ cells were also multilayered (but not aggregated) in the *Exoc1* cKO (Author response image 2, white arrowhead). For these reasons, we consider that the reason for the appearance of such a multilayered cell population is not that the undifferentiated state is disorganized, but rather that the physical pressure is reduced by the emptying of the lumen of the seminiferous tubule. Since this is just a hypothesis and there is no evidence for it, we would like to analyze it in the future.

**Author response image 2. respfig2:** 

Also, as fewer differentiation-primed cells are generated, the numbers of differentiating spermatogonia should be reduced in the knockout. Is this the case?

Yes, it is. I present this data in Figure 5—figure supplement 1 and describe in Lines 333-335 in the main text.

14. The manuscript text should be checked carefully for accuracy throughout. For instance, in the introduction, second paragraph, spermatogonia are stated to be divided into undifferentiated, differentiation-primed and differentiated fractions. However, differentiation -primed cells are found within the undifferentiated population as are SSCs. The significance of markers used for different spermatogonial populations (e.g. GFRa1) should be included in the introduction for clarity. The discussion should be modified to account for new datasets and models.

Following your advice, I have changed these notations to Lines 75-86, 295-298, 325-327, 386, 390, 417-421.

15. Many datasets are not quantified (see specific points above). Statistics cannot be evaluated as for most experimental settings the basic information (N number, means and coefficient of Variation… ) is not provided.

As I replied in Essential revisions comment 3, I have included the quantified data. The number of analyses and other details are described in each figure legends.

16. Additionally, the actual study design is not described. Therefore, the validity of the findings is partially questionable. No Information is provided on the number of mice analyzed. Nature and size of experimental groups and controls are not mentioned. Strategies for histological analysis are poorly described. Did the authors use random systematic sampling approaches on sufficient number of samples? The excellent micrographs must be considered individual observations as generalization of the descriptions would only be valid if repetitions are reported. Similar concerns apply to the blotting results. To check for validity and statistically Sound analysis, the authors need to provide a flow chart or table of all experimental animals and the endpoints analyzed in specific groups.

As I replied in Essential revisions comment 3, I have included the quantified data. The number of analyses and other details are described in each figure legends. The method of image analysis was described in Lines 626-640.

17. The mouse models are not as well defined as the authors Claim. Snap23 k/O mice Show germ cell development up to spermatids (Figure 3). Can the authors elaborate more on the presence of multinucleated cells and spermatids at the same time? Was that stable in all animals of this genotype?

As I replied in Essential revisions comment 3, the analysis frequency of AGS in *Snap23* cKO was added to Figure 3—figure supplement 2.

[Editors' note: further revisions were suggested prior to acceptance, as described below.]

The manuscript has been significantly improved but there are some remaining issues (many could be amended with textual revisions) that need to be addressed, as outlined below:1. Fluorescent images are still not clear.

We revised Figure 1C, Figure 2A, 2F, 2G and Figure 5A, which are fluorescence images.

2. Line 161: The authors mentioned "EXOC1 is observed in every cell in the adult testes" in the Figure 1 legend but described "EXOC1-PA was also detected in all male germ cells observed" in text. They may want to focus on germ cells, but it's better clarifying both Sertoli and germ cells are positive in the text if all cells are stained in testicular tubules.The authors could not judge if Sertoli cells are positive with the current figure.

Thank you for your suggestion. We have corrected lines 167-172 and 893-895 as per your suggestion.

Lines 167-172

Before the revision:

“while no signal could be detected in the *Exoc1^PA-N/PA-N^* mice using any method. These data indicate that EXOC1 protein is expressed in male mouse germ cells.”

After the revision:

“while no signal could be detected in the *Exoc1^PA-N/PA-N^* mice using any method. **In *Exoc1^+/PA-C^* adult mice, PA signals were also detected in Sertoli cells, which are located at *the basal compartment of the* seminiferous tubules and whose nucleus are euchromatic with a large nucleolus (Franca et al., 2016). These data indicate that EXOC1 protein is expressed in male mouse germ and Sertoli cells.”**

Line 893-895

Before the revision:

“EXOC1 is observed in every cell in the adult testes.”

After the revision:

“EXOC1 is observed in every cell in the adult testes. **The arrowheads indicate Sertoli cells in which the nucleus is eurochromatin with a large nucleolus.”**

3. It was originally requested for the authors to assess the efficiency of floxed gene knockout in their models. While they have assessed whether remaining sperm in the Exoc1 conditional KO model are gene deleted, analysis of testis from this and other KO models described has not been performed. A simple qRT-PCR on total testis extracts or isolated testis cells could be used. It is important to define efficiency of gene knockout as the phenotype of some of the models is mild, e.g. Snap23 KO.

Thank you for your suggestion. This is due to our insufficient preparation, but we do not have live cKO mice, frozen samples, and mRNA stock, so we need more than 6 months to prepare for the experiments you kindly suggested. Although we thought your point was plausible, we believe that this RT-qPCR data is not necessarily needed in this study. The reasons for this are as follows.

EXOC1 has been shown to be expressed in spermatocytes and later stages of differentiation. These differentiated cells comprise the majority of all cells in the testis. Therefore, the fraction of each cell population that constitutes the testis drastically altered between the *Exoc1* cKO and wild-type. In this situation, if RT-qPCR is performed with entire testis, it is not possible to determine whether the difference in *Exoc1* expression between the cKO and wild-type is due to the deletion of the gene or to the alteration in the fraction of the germ cell population. A further complication is that the transcripts are shared among the connected cells that comprise the syncytium. If recombination occurs after A_pair_ but not A_single_, there might be a syncytia with a dilute gene dosage of *Exoc1* or *Snap23*. Moreover, although we cannot completely clarify whether it is complete or incomplete, we could well claim that the loss of *Exoc1* leads to abnormal pseudopodia formation of spermatogonia and aggregation of spermatocytes in many cases based on the data that we have presented so far (such as the data on the counting of seminiferous tubules).

The results of Figure 3D and Figure 3—figure supplement 2B clearly showed that *Snap23* was involved in the formation or maintenance of spermatocyte syncytia. Since the phenotyping of the *Snap23* cKO was performed under almost the same conditions as that of the *Exoc1* cKO (same Cre driver, same age at analysis, almost same floxed distance, and same genetic background), the requirement of *Snap23* for the formation or maintenance of the syncytium may be lower than that of *Exoc1*. However, since we do not have experimental data to deny the possibility that the recombination efficiency of *Exoc1* cKO and *Snap23* cKO may differ, we have revised the text as follows to include both possibilities.

Line 270-279

Before the revision:

“This suggests that *Snap23* is dispensable for spermatogenesis and that another protein in the SNAP family could be compensating for that function. […] These results suggest that EXOC1 regulates the formation of the correct syncytium structure in cooperation with STX2 and SNAP23.

After the revision:

“This suggests that *Snap23* is dispensable for spermatogenesis and that another protein in the SNAP family could be compensating for that function. **[…]** These results suggest that EXOC1 regulates the formation of the correct syncytium structure in cooperation with STX2 and SNAP23.”

We think that it is very important to know how often the deletion of *Exoc1*/*Snap23* affects each phenotype, and to do so, we need to understand the efficiency of Cre-loxP recombination as you pointed out. However, to evaluate the magnitude of the impact of the loss of those genes, it is also essential to establish the challenging experimental techniques to analyze aggregated spermatocytes perfectly in 3D and the degree of *Exoc1*/*Snap23* mRNA sharing in syncytium consisting of non-recombinant and recombinant cells. These issues are very fascinating and intriguing, but we think they are the next challenge.

4. A new N-terminally PA tagged Exoc1 model is described in the revision as the C-tagged version clearly disrupts EXOC1 function, which is embryonic lethal. Confusingly, while the homozygous N-tagged mice are viable, indicating that EXOC1 function is retained, the authors could not detect expression of the N-tagged EXOC1 by immunostaining and suggest that it might be unstable. This seems counterintuitive and text should be modified to help resolve and discuss this issue. On a related point, lines 160-163 where rationale for use of the LG3 tag in the fusion construct is described – this should be reworded for clarity.

Thank you very much for pointing this out. I have followed your suggestion and corrected lines 158-164.

Before the revision:

“The signal intensity of the band, on the western blot, was lower for *Exoc1^+/PA-N^* than that for *Exoc1^+/PA-C^*, which might be because EXOC1-PA-N is more easily decomposed.”

After revision:

“The signal intensity of the band, on the western blot, was lower for *Exoc1^+/PA-N^* than that for *Exoc1^+/PA-C^*. […] Since the *Exoc1^PA-N^* homozygous mutant, unlike the *Exoc1^PA-C^* homozygous mutant, did not show a pronounced abnormal phenotype, we considered that EXOC1-PA-N is probably more similar in

behavior and function to the wild-type EXOC1 than EXOC1-PA-C.”

5. It is stated that while GFRa1+ spermatogonial pseudopods in the Exoc1 KO are shorter, those of the Stx2 KO are not, suggesting that EXOC1 operates independently of STX2 in this context. However, the pseudopods of Stx2 KO are strongly trending to be shorter and P value is quoted as 0.052. This suggests that a larger dataset may indeed show significant reduction and the conclusions from this data are overstated. Text should be changed appropriately to reflect this strong trend in data and/or more cells quantified to assess the robustness of the conclusions.

Thank you very much for your valuable comments. We were preoccupied with our preconceived notions and were not able to make an accurate assessment. As you pointed out, we also think the Stx2 KO may well have a shorter pseudopod. Therefore, we have revised lines 299-307, lines 943-946.

Lines 299-307:

Before the revision:

“In the *Stx2* KO mice generated in this study, as in the *Stx2^repro34^* mice (Fujiwara et al., 2013), spermatocytes were observed to aggregate (Figure 4—figure supplement 1C), but no abnormalities in the length of the spermatogonia pseudopod of the *Stx2* KO adult mice were observed (Figure 4A). […] These results suggest that EXOC1 functions in the pseudopod elongation of GFRα1^+^ spermatogonia independently of STX2.”

After the revision:

“In the *Stx2* KO mice generated in this study, as in the *Stx2^repro34^* mice (Fujiwara et al., 2013), spermatocytes were observed to aggregate (Figure 4—figure supplement 1C).[…] These results suggest that EXOC1 functions in the pseudopod elongation of GFRα1^+^ spermatogonia partially dependently, but not completely dependently of STX2.”

Lines 943-946

Before the revision:

“(B) Pseudopod length quantification using sections. Average length of GFRα1^+^ spermatogonia pseudopods in *Exoc1* cKO was shorter than that of control (n = 3 in each genotype, 23–25 cells in each mouse). […] (n = 3 in each genotype, 40–44 cells in each mouse, *p = 0.052, student’s t-test).”

After the revision:

“(B) Pseudopod length quantification using sections. Average length of GFRα1^+^ spermatogonia pseudopods in *Exoc1* cKO was shorter than that of *Stx2* KO and wild type (n = 3 in each genotype, 25–36 cells in each mouse). *p = 0.052, **p = 1.8 × 10^-6^, ***p = 9.5 × 10^-9^. one-way ANOVA.”

6. The abstract could be reworded for clarity. A sentence about role of EXOC1, SNAP23 etc in the exocyst complex should be included up-front to help non experts in the field. The term "high regulation" in the first sentence is ambiguous.

We have revised the abstract according to your recommendations. Due to word count limitations, we were unable to describe the functions of these proteins, but we have tried to mention the families to which they belong.